

# Passive ground-based remote sensing of radiation fog

Heather Guy[1,2], David D. Turner[3], Von P. Walden[4], Ian M. Brooks[2], and Ryan R. Neely[1,2]

[1]National Centre for Atmospheric Science, Leeds, U.K.
[2]School of Earth and Environment, University of Leeds, Leeds, U.K.
[3]Global Systems Laboratory, National Oceanic and Atmospheric Administration, Boulder, CO, USA
[4]Department of Civil and Environmental Engineering, Laboratory for Atmospheric Research, Washington State University, Pullman, WA, USA

**Correspondence:** Heather Guy (heather.guy@ncas.ac.uk)

**Abstract.** Accurate boundary layer temperature and humidity profiles are crucial for successful forecasting of fog, and accurate retrievals of liquid water path are important for understanding the climatological significance of fog. Passive ground-based remote sensing systems such as microwave radiometers (MWRs) that have multiple channels between 22-31 GHz and 51-59 GHz, and infrared spectrometers such as the Atmospheric Emitted Radiance Interferometer (AERI) which measures spectrally resolved infrared radiation (3.3 to 19.2 $\mu m$) can retrieve both thermodynamic profiles and liquid water path. Both instruments are capable of long-term unattended operation and have the potential to support operational forecasting. Here we compare physical retrievals of boundary layer thermodynamic profiles and liquid water path during 13 cases of supercooled radiation fog from a MWR and an AERI collocated in central Greenland. We compare both sets of retrievals to in-situ measurements from radiosondes and surface-based temperature and humidity sensors. The retrievals based on AERI observations accurately capture shallow surface-based temperature inversions (0-10 m a.g.l) with lapse rates of up to -1.2°C m$^{-1}$, whereas the strength of the surface-based temperature inversions retrieved from MWR observations are uncorrelated with in-situ measurements. For all but one case study, the retrievals based on AERI observations detect fog onset (defined by a threshold in liquid water path) earlier than those based on MWR observations by up to 4 hours. We propose that due to the high sensitivity of the AERI instrument to near-surface temperature and small changes in liquid water path, the AERI (or an equivalent infrared spectrometer) could be a useful instrument for improving fog monitoring and nowcasting.

## 1 Introduction

The socioeconomic and climatological impacts of fog are far-reaching. The reduction in visibility associated with fog disrupts transportation, resulting in economic losses equivalent to those associated with tornadoes and severe storms (Gultepe et al., 2007). Poor visibility due to fog is the most impactful extreme weather event in Arctic maritime operations (Panahi et al., 2020) and the second largest contributor to weather related accidents in aviation after adverse winds (Gultepe et al., 2019). Supercooled fog is particularly impactful since the collision of supercooled liquid droplets with a cold surface can result in the formation of rime or glaze ice. The build-up of ice can damage structures and power transmission lines (Ducloux and Nygaard, 2018) and presents an additional safety hazard in both shipping and aviation (Cao et al., 2018; Panahi et al., 2020),





making accurate forecasts of supercooled fog critical for risk mitigation. From a climatological perspective, fog is an important

moisture source, particularly in arid regions, (e.g. Hachfeld et al., 2000), and impacts the surface energy budget by modifying radiant and turbulent energy transfer (Shupe and Intrieri, 2004; Beiderwieden et al., 2007; Anber et al., 2015). The hydrological and radiative impacts of fog are both directly related to fog duration and liquid water content, and so accurate monitoring of fog liquid water content is vital for understanding the role of fog in local climate and hydrological cycles.

Fog forms when the near-surface air reaches saturation resulting in the formation of liquid water droplets on condensation

nuclei (e.g. Oke, 2002). The air can reach saturation either through cooling until it reaches the dew point or through a moistening process such as the evaporation of surface water/drizzle or moist air advection (Gultepe et al., 2007). The cooling of air near the surface can result from advection (either cold air advection or the advection of a warm air mass over a cooler surface), through orographic effects (i.e. the adiabatic cooling of air rising over topography, or cold air pooling in valleys) or through direct radiative cooling of the surface. Fogs that primarily form through radiative cooling of the surface are known as radiation

fogs and commonly form on clear evenings with light winds - where the net surface cooling is maximised through the reduction of direct solar heating and limited turbulent mixing of heat downward to the surface (e.g. Savijärvi, 2006). Due to the rapid cooling of the surface, the formation of radiation fog is associated with a surface temperature inversion, which can be extremely shallow, with most of the inversion often developing in the lowest 10 m above the surface (Hudson and Brandt, 2005; Price, 2011; Izett et al., 2019).

The onset of radiation fog in numerical weather prediction (NWP) models is particularly sensitive to the initial thermodynamic structure of the boundary layer (Steeneveld et al., 2014). Accurate representation of boundary layer structure, particularly temperature and humidity profiles in the lowest 1 km a.g.l and the development of the surface-based temperature inversion, is thus crucial for forecasting radiation fog (Steeneveld et al., 2014; Gultepe et al., 2007; Bergot et al., 2007; Tardif, 2007); however, NWP models often fail to reproduce the strong but shallow gradients associated with it (Martinet et al., 2020; Westerhuis

and Fuhrer, 2021).

The assimilation of boundary layer thermodynamic profile measurements is one possibility for improving NWP forecasts of radiation fog. However, making continuous high-resolution observations of temperature and humidity profiles is challenging. Despite improvements in recent years, satellite retrievals of boundary layer profiles and fog characteristics remain insufficient due to their coarse vertical resolution (> 1 km) and poor spatial coverage (Wulfmeyer et al., 2015; Wu et al., 2015; Wilcox,

2017; Yi et al., 2019). Surface-based in-situ measurements are limited by a maximum height (usually less than 50 m), while radiosonde profiles are spatially and temporally sparse and resource intensive, and the development of a coordinated unmanned aerial system profiling platform is still in its infancy (Jacob et al., 2018; McFarquhar et al., 2020). Active ground-based remote sensors, such as differential absorption lidars (DIALs), can produce accurate thermodynamic profiles with a high temporal resolution, but have a typical lowest range gate of greater than 100 m, making them unsuitable for fog monitoring (Newsom

et al., 2020; Stillwell et al., 2020; Turner and Lohnert, 2021).

In addition to thermodynamic profiles, accurate monitoring of liquid water content is important to understand the climatological and hydrological impacts of fog. One metric to describe the liquid water content is fog liquid water path (LWP), defined as the integral of liquid water content over the depth of the fog layer. LWP is directly related to visibility; for example, given a


homogeneous, mono-disperse fog with a depth of 100 m and a uniform droplet effective radius of 10 $\mu$ m, increasing the LWP

from 10 g m$^{-2}$ to 20 g m$^{-2}$ corresponds to a reduction in horizontal visibility from 200 to 100 m (assuming a visible contrast threshold of 0.05; Bendix, 1995), highlighting the importance of accurate LWP retrievals for visibility nowcasting. The LWP of thin fogs (LWP < 40 g m$^{-2}$) is important from a climatological perspective because both longwave and shortwave surface radiative fluxes become extremely sensitive to small changes in LWP (Turner et al., 2007a). Although thin liquid clouds and fogs are common globally (Turner et al., 2007a), they are especially important in the Arctic where they dominate cloud radia-

tive forcing of the surface (Shupe and Intrieri, 2004; Miller et al., 2015). Cloud LWP was a critical control on the exceptional Greenland Ice Sheet melt event of 2012; at the highest point on the ice sheet, a change in cloud LWP by $\pm$ 20 g m$^{-2}$ from the observed value of $\sim$ 25 g m$^{-2}$ would have been sufficient to prevent surface melt (Bennartz et al., 2013).

    Ground-based microwave radiometers (MWRs) are passive sensors that typically measure downwelling radiation at 22-31 GHz and 51-58 GHz in 14 to 35 different spectral channels and are sensitive to the temperature and water vapour profile in the

lowest 6 km of the atmosphere (Löhnert and Maier, 2012; Blumberg et al., 2015). Because MWRs can retrieve continuous (< 10 s) boundary layer temperature and humidity profiles as well as LWP under both clear skies and non-precipitating clouds, they are frequently used for fog monitoring (e.g. Gultepe et al., 2009; Wærsted et al., 2017; Temimi et al., 2020; Martinet et al., 2020). Recent studies have demonstrated that the assimilation of MWR brightness temperatures into NWP models has the potential to improve forecasts of stable boundary layers and fog by correcting errors in the temperature profile in the lowest

500 m above the surface (Martinet et al., 2017, 2020). The success of these trials contributed to EUMETNET's recent decision to establish a homogeneous European network of MWRs by 2023 (Illingworth et al., 2019; Rüfenacht et al., 2021).

    Despite the promise of MWRs to improve fog forecasts, the maximum vertical resolution of boundary layer temperature profile retrievals (50 m at the surface decreasing to 1.7 km at 1 km a.g.l; Rose et al., 2005; Cadeddu et al., 2013) is insufficient to resolve the shallow surface-based temperature inversions that often portend the onset of radiation fog (Price, 2011; Izett et al.,

2019). Although combining the MWR with active remote sensing instruments such as DIALs or Radio Acoustic Sounding Systems (RASS) can improve the vertical resolution of the temperature profile retrievals in the lowest 2 km of the atmosphere, these improvements do not extend down to the lowest 100 m a.g.l due to the height of the lowest range gate of the active remote sensing instruments (Turner and Lohnert, 2021; Djalalova et al., 2021). In addition, large absolute uncertainties in LWP retrievals from the MWR ($\pm$ 12-25 g m$^{-2}$) result in large errors during thin fog (LWP < 40 g m$^{-2}$; Turner, 2007b).

Another passive remote sensing instrument, the Atmospheric Emitted Radiance Interferometer (AERI, Knuteson et al., 2004a) has greater sensitivity than MWRs to both changes in near surface (< 1 km) thermodynamic profiles (Blumberg et al., 2015; Turner and Lohnert, 2021) and small changes in LWP (for LWP < 40 g m$^{-2}$, Turner, 2007b). The AERI measures spectrally resolved downwelling infrared radiation between 3.3 and 19.2 $\mu$m. Because of the higher opacity at infrared wavelengths relative to the optical depths spanned by the MWR, the AERI can detect changes in the boundary layer thermodynamic profile

at a finer vertical resolution and with greater accuracy than the MWR (Blumberg et al., 2015; Turner and Lohnert, 2021). The primary disadvantage is that the AERI is not sensitive to atmospheric properties above a cloud with a LWP > $\sim$ 40 g m$^{-2}$, for which the cloud is nearly opaque in the infrared. This means that retrievals of thermodynamic profiles above optically thick clouds are not possible, and retrievals below them are only possible if the cloud temperature and height are well characterised.



Although several studies have compared the performance of AERI and MWR retrievals of thermodynamic profiles and LWP
under different conditions (Blumberg et al., 2015; Turner, 2007b; Löhnert et al., 2009; Turner and Lohnert, 2021), none of
these studies have included cases of fog. Fog is distinct from 'cloudy scenes' in general because the LWP and changes in the
thermodynamic profile that are relevant for fog development and life-time are concentrated in the lowest layers (< 100 m)
above the surface. The goal of this study is to compare the performance of thermodynamic and LWP retrievals based on MWR
and AERI observations during fog events, with an emphasis on those aspects that are crucial for making accurate forecasts
and understanding the climatic impact of fog: The representation of the thermodynamic profile in the lowest 1 km a.g.l, the
detection of shallow surface based temperature inversions, and accurate measurements of small changes in fog LWP.

We take advantage of the collocation of a MWR and an AERI alongside a large suite of supplementary instruments for
monitoring atmospheric properties at Summit Station (Summit), in the centre of the Greenland Ice Sheet (Shupe et al., 2013).
The surface air temperature at Summit approaches 0°C only in exceptional circumstances (NSIDC, 2021), and supercooled
radiation fog is common, occurring over 10% of the time in the summer (Cox et al., 2019). Summer-time radiation fog in
central Greenland is particularly impactful because it forms during the coldest part of the day and has a net warming effect
at the surface, effectively dampening the diurnal temperature cycle with the potential to precondition the ice sheet surface for
melt (Solomon et al., 2017; Cox et al., 2019). Aviation operations at Summit are also frequently disrupted by the low visibility.

Using a consistent physical retrieval algorithm for both instruments, we compare the suitability of the MWR and the AERI
for retrieving near surface thermodynamic profiles and LWP during super-cooled radiation fog events at Summit in the summer
of 2019. We evaluate the retrieved thermodynamic profiles against radiosonde profiles and in-situ temperature and humidity
measurements and assess the ability of each set of retrievals to detect the increase in LWP associated with the onset of fog.
Henceforth in this study 'fog' will specifically pertain to supercooled radiation fog unless otherwise specified.

## 2  Methods

### 2.1  Measurement site and instrumentation

The Integrated Characterization of Energy, Clouds, Atmospheric state and Precipitation at Summit (ICECAPS) project col-
lected continuous observations of the atmosphere above Summit from 2010 to 2021 (Shupe et al., 2013). At the highest point
of the Greenland Ice Sheet (-38.45 E, 72.58 N, 3250 m a.s.l.), the atmosphere above Summit is extremely dry and temperatures
are rarely above freezing (Shupe et al., 2013). The ice sheet surface is homogeneous in all directions, so that the atmospheric
conditions at Summit are minimally influenced by local topography. During the summer (JJAS), freezing fog (defined as fog
that reduces visibility to less than 1,000 m) was reported by on-site observers 10 % of the time (2010-2020). These fogs can
occur when surface temperatures are as low as -35°C and almost always contain supercooled liquid droplets (Cox et al., 2019),
presumably due to a lack of ice nucleating particles (near surface aerosol concentrations at Summit are exceptionally low, Guy
et al., 2021). The applicability of the results of this study to other (less extreme) environments is discussed in section 4.

The Aerosol Cloud Experiment (ACE) was added to the ICECAPS project in 2019 and included the addition of temperature
and humidity sensors and sonic anemometers at four levels on a 15 m tower for high resolution monitoring of the near-surface





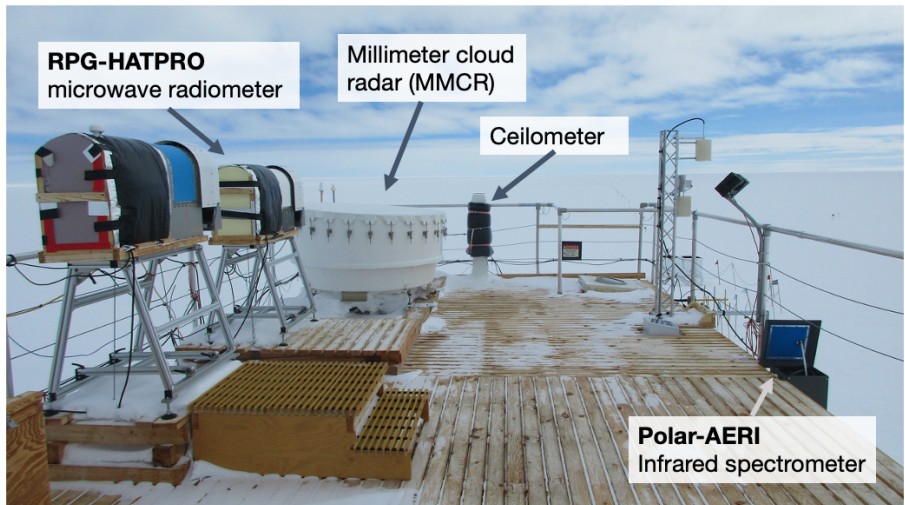

**Figure 1.** Key ICECAPS-ACE instrumentation at Summit Station (photographed by the author, 16 May 2019).

turbulent and thermodynamic structure (Guy et al., 2021). For this study, we focus on fog events during the summer of 2019 whilst the multi-level temperature and humidity data from the tower are available. Figure 1 shows the experimental setup of the MWR and AERI at Summit, and Table 1 provides details of all the ICECAPS-ACE instrumentation used in this study.

We use the tower mounted temperature and humidity probes (Vaisala HMP155, installed in aspirated shields) as 'true' reference points to assess the performance of the surface temperature retrievals from the MWR and AERI. The instrument uncertainty for the HMP155 is $< \pm 2.5\%$ relative humidity and $< \pm 0.3°$C. For this study, we define the 'surface' as the height of the raised platform photographed in Fig. 1, where both the AERI and MWR windows are situated. This surface is aligned with the HMP155 sensor mounted on the tower at 4 m above the snow surface; measurements from this sensor are henceforth

referred to as surface temperature and humidity. Measurements from the HMP155 sensor located 10 m higher on the tower are compared to the 10 m thermodynamic retrievals. The same height adjustment is applied to the radiosonde profiles prior to comparison (which are launched approximately 3 m below the platform 'surface', or 1 m above the snow surface). The uncertainty in the radiosonde measurements is $\pm 4\%$ relative humidity and $\pm 0.3°$C (Jensen et al., 2016).

### 2.1.1    The AERI

The polar AERI (PAERI) at Summit was designed and manufactured by personnel at the Space Science and Engineering Center (SSEC) at the University of Wisconsin-Madison, and is one of the original AERIs developed for the US Department of Energy's Atmospheric Radiation Measurement (ARM) program (Turner et al., 2016). It was built and calibrated according to specifications in Knuteson et al. (2004a) and adheres to the performance requirements of radiometric calibration ($< 1\%$, 3 $\sigma$, of ambient radiance) and spectral calibration (1.5 ppm, 1 $\sigma$) as explained by Knuteson et al. (2004b). The PAERI measures

downwelling spectral infrared radiance between 3 and 19 $\mu$m at an unapodized spectral resolution of about 0.48 cm$^{-1}$ (see



**Table 1.** Overview of instrumentation used in this study. All instruments were installed at Summit as part of the ICECAPS project (Shupe et al., 2013) or ICECAPS-ACE project (Guy et al., 2021).

| Title | Instrument | Key specifications | References |
|---|---|---|---|
| PAERI | Polar Atmospheric Emitted Radiance Interferometer | 530-3,000 cm$^{-1}$ (3-19 $\mu$m), 1 cm$^{-1}$ res. < 1-min time res. | (Knuteson et al., 2004a) (Walden et al., 2005) |
| HATPRO MWR | RGP Humidity and Temperature Profiler, microwave radiometer | Frequencies: 7 channels 22-32 GHz, 7 channels 51-58 GHz, 2-4 s time resolution. | (Rose et al., 2005) |
| MMCR | Millimetre cloud radar | Ka band (35 GHz), 8-mm wavelength, 45 m vertical res. 2 s time res. | (Moran et al., 1998) |
| Ceilometer | Vaisala laser ceilometer CT25K | 905 nm wavelength. 15 m vertical res., 15 s time res. | (Münkel et al., 2006) |
| POSS | Precipitation Occurrence Sensor System | X-band radar (10.5 GHz) 1-min time res. Single volume near surface. | (Sheppard and Joe, 2008) |
| Temperature and humidity probes | Vaisala HMP155, aspirated | 1-min averages at 0 m and 10 m. | (Guy et al., 2020) |
| Radiosondes | Vaisala RS41-SG | Launched at 12 and 00 UTC daily. | (Jensen et al., 2016) |

Table 3 from Knuteson et al., 2004a). The PAERI operates on a continuous measurement schedule where it obtains views of the hot and ambient calibration sources, followed by eight consecutive views of the sky at zenith. The sequence is repeated so that each set of eight sky views is bracketed in time by views of both calibration sources; these sources are then used to calibrate the eight sky views. Each of the spectral measurements is a 'co-addition' of six interferometric scans. Each complete, calibrated

measurement sequence takes approximately 3 minutes with the sky views separated by approximately 15-20 s. This yields more than 3300 infrared spectra each day. Quality control is then applied to each of the spectra by eliminating those that have instrument parameters outside of acceptable limits; the acceptable limits were set by SSEC personnel. The important instrument parameters are the responsivities and noise-equivalent radiances of the hot blackbody calibration source measured by both of the PAERI detectors (InSb and MCT), plus the current and temperature of the Stirling cooler that maintains the detectors at 77

K. In actuality, the instrument responsivities are a very sensitive indicator of the PAERI's health, and most unusable spectra are eliminated by low responsivity associated with small amounts of snow on the PAERI scene mirror. Finally, the remaining





calibrated sky views are subjected to noise filtering using the technique described by Antonelli et al. (2004) and Turner et al. (2006).

### 2.1.2 The MWR

The MWR at Summit, a Humidity and Temperature Profiler (HATPRO) from Radiometer Physics GmbH, observed downwelling radiation in all 14 channels simultaneously every 4 seconds (see table 2 for channel details). After collecting 600 zenith views, the HATPRO collected elevation scans at 5.4, 10.2, 16.2, 19.2, 23.4, 30.0, and 42.0 on either side of zenith. These elevation scans were used to both evaluate and update the calibration accuracy of the K-band channels (i.e., the low opacity channels between 22 and 32 GHz) using the tip-curve technique (Han, 2000). The more opaque V-band channels (i.e., in the 51

to 58 GHz band) were calibrated twice yearly using an external liquid nitrogen target; the most recent calibration used for this analysis was performed on 01 May 2019. Both the tip curves and the liquid nitrogen views are used to determine the effective temperature of the internal noise diode, which is used regularly when viewing the internal blackbody to establish two different reference values (i.e., one ambient blackbody view with the noise diode off, and one 'hot' blackbody view with the noise diode on). These internal blackbody views, which are done every minute, are used to continually update the gain of the radiometer

and convert the observed signal to brightness temperature, following the calibration principles outlined in Liljegren (2000).

However, since the liquid nitrogen calibrations are performed infrequently, any drift in the effective temperature of the noise diode in the V-band channels will result in a calibration bias. Using a radiative transfer model (the monochromatic MonoRTM, Clough et al., 2005) with radiosonde profiles as input, we have determined a brightness temperature offset that is subtracted from the observed brightness temperatures. The bias correction, and the impact of not applying this correction prior

to performing the thermodynamic retrievals are discussed in Appendix A.

### 2.2 Case study identification

To identify case studies of radiation fog under otherwise clear skies, we only considered times when there were no clouds detected by the MMCR, which has a lowest range gate close to 200 m a.g.l and is therefore insensitive to fog, and when there was no precipitation detected by the POSS. Of the times that met these criteria, fog was provisionally identified when

the 962 cm$^{-1}$ downwelling radiance measured by the AERI was greater than a threshold of 1.7 RU (radiance units, 1RU = 1 mW m$^{-2}$ sr$^{-1}$ cm$^{-1}$). In the extremely dry atmosphere over Summit, clear-sky transmittance is almost unity at 962 cm$^{-1}$, so this micro-window is particularly sensitive to the presence of clouds (e.g. Cox et al., 2012). The threshold value of 1.7 RU is three standard deviations above the mean 962 cm$^{-1}$ radiance during 236 verified clear sky hours between June and Sept 2019, and therefore identifies when the AERI window was obscured by cloud/fog. Ambiguous cases when there was evidence that

something other than fog may have caused the 962 cm$^{-1}$ radiance increase, such as clear sky ice crystal precipitation, high cirrus clouds, or the plume from the station generator, were removed based on the observer log and photographs. Table 3 details the 13 cases that met the criteria above and were selected for the intercomparison. In each case, the fog forms in late evening or early morning and usually dissipates by midday as is characteristic of radiation fog (Fig. 2).



**Table 2.** Centre frequencies, assumed noise level, whether the elevation scans for the frequency are included in the observation vector, and the bias offset applied to the observations for the HATPRO MWR at Summit.

| Frequency GHz | Noise Level K | Used in Elev. Scan | Bias Offset K |
|---|---|---|---|
| 22.24 | 0.30 | No | 0.23 |
| 23.04 | 0.30 | No | 0.08 |
| 23.84 | 0.30 | No | 0.09 |
| 25.44 | 0.30 | No | 0.00 |
| 26.24 | 0.30 | No | 0.12 |
| 27.84 | 0.30 | No | 0.17 |
| 31.40 | 0.30 | No | 0.15 |
| 51.26 | 0.80 | No | 2.02 |
| 52.28 | 0.80 | No | 2.15 |
| 53.86 | 0.50 | No | 2.03 |
| 54.94 | 0.30 | Yes | -0.45 |
| 56.66 | 0.30 | Yes | -0.32 |
| 57.30 | 0.25 | Yes | -0.12 |
| 58.00 | 0.25 | Yes | -0.11 |

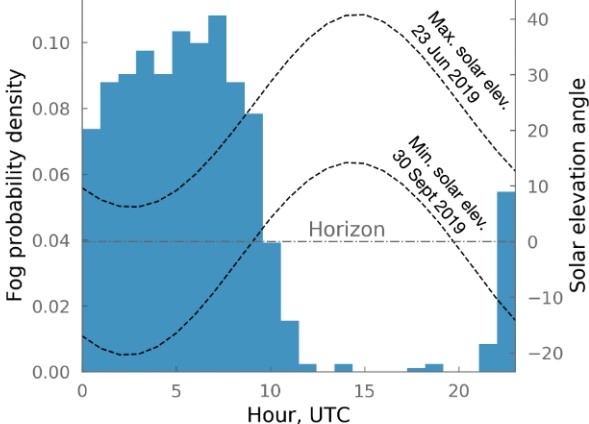

**Figure 2.** Diurnal distribution of fog during the summer 2019 case studies listed in Table 3 (blue bars). Black dashed lines show the maximum and minimum solar elevation angles (Jun – Sept 2019). Local time at Summit is UTC-3 h.


**Table 3.** Details of the 13 radiation fog cases used in this study including mean temperatures (T) and water vapor mixing ratios (wv). Note that the minimum visibility comes from observer reports at 00, 12 and 18 UTC and may not represent the minimum visibility outside of these times. A value of 9999 is reported if horizontal visibility is not impeded. Local time is UTC-3h.

| ID | Case start Date Time. UTC, 2019 | Case end Date Time UTC, 2019 | Duration (h) | Mean surface T (°C) | Mean surface wv (g kg$^{-1}$) | Min visibility (observer log, m) |
|----|------|------|------|------|------|------|
| 1  | 08 Jun 03:30 | 08 Jun 10:00 | 6.5 | -17 | 1.3 | 4,800 |
| 2  | 12 Jun 02:00 | 12 Jun 10:30 | 8.5 | -9.6 | 2.5 | 4,800 |
| 3  | 13 Jul 22:00 | 14 Jul 07:00 | 9.0 | -20 | 0.96 | 1,600 |
| 4  | 15 Jul 21:00 | 16 Jul 11:00 | 14 | -18 | 1.2 | 400 |
| 5  | 31 Jul 22:25 | 01 Aug 08:30 | 10 | -10 | 2.4 | 400 |
| 6  | 01 Aug 18:00 | 02 Aug 15:00 | 21 | -10 | 2.3 | 800 |
| 7  | 03 Aug 19:00 | 04 Aug 09:00 | 14 | -14 | 1.7 | 9999 |
| 8  | 04 Aug 21:30 | 05 Aug 12:00 | 15 | -17 | 1.3 | 400 |
| 9  | 05 Aug 23:00 | 06 Aug 10:00 | 11 | -20 | 0.91 | 1,600 |
| 10 | 09 Aug 19:00 | 10 Aug 04:00 | 9.0 | -16 | 1.4 | 1,600 |
| 11 | 14 Aug 23:00 | 15 Aug 10:00 | 11 | -26 | 0.54 | 3,200 |
| 12 | 05 Sep 02:00 | 05 Sep 12:00 | 10 | -25 | 0.59 | 400 |
| 13 | 30 Sep 02:00 | 30 Sep 12:00 | 10 | -27 | 0.47 | 4,800 |

## 2.3 Retrieval methodology

We retrieve boundary layer thermodynamic profiles (temperature, T, and water vapor mixing ratio, wv) and LWP at a 5-min temporal resolution using the TROPoe iterative optimal estimation physical retrieval algorithm that is detailed in Turner and Lohnert (2021) and Turner and Blumberg (2019). TROPoe uses a forward model to calculate the observation vector from the current state vector, where the state vector is the retrieved thermodynamic profile and LWP, and the observation vector is the downwelling radiance observed by either the polar AERI or HATPRO MWR. Surface meteorological measurements are not

included in the observation vector to allow for their use as an independent validation measurement; the impact of including surface measurements in the retrievals is discussed in Appendix B. Note that the observation vector from the MWR includes data from the elevation scans at 10.2, 16.2, and 19.2 degrees for the four most opaque V-band channels; including elevation scans in the retrieval has been shown to increase the accuracy of the retrieved temperature profile (Crewell and Lohnert, 2007). The forward models are line-by-line radiative transfer models; the LBLRTM version 12.1 (Clough and Iacono, 1995) simulates

the AERI spectral radiances, and the monochromatic MonoRTM (Clough et al., 2005) simulates the MWR radiances. Note that the latter uses the improved temperature-dependent liquid water absorption coefficients (Turner et al., 2016). The state vector is incrementally adjusted to minimise the difference between the forward model calculation and the observation vector until the





change between successive iterations is less than the uncertainty in the current state vector (Rodgers, 2000). We run TROPoe in two physically consistent configurations; once using only the PAERI radiances as the observation vector (as in Turner and Löhnert, 2014, henceforth named AERIoe) and once using only the microwave brightness temperature observations from the HATPRO MWR (as in Löhnert et al., 2009, henceforth named MWRoe).

Thermodynamic retrieval from passive spectral radiance observations is an ill-posed problem, hence the optimal-estimation retrieval is necessarily constrained by an a priori probability density function (the prior) that provides the first guess state vector that stabilises the retrieval (Turner and Löhnert, 2014). Typically, a location specific prior can be derived from a database of historical observations (i.e. from radiosonde profiles) at or near the location of interest. The prior for Summit is computed from 1756 summer radiosonde launches (2010-2018). However, due to the rapid warming in the Arctic (e.g. Koenigk et al., 2020), this does not encapsulate the exceptionally warm and moist conditions at Summit during the summer of 2019 (NSIDC, 2019). To allow the retrievals more flexibility to account for the exceptional conditions, we have re-centered the prior using the mean of the three closest radiosondes to the retrieval date (whilst conserving relative humidity), and increased the wv variance in the prior by a factor of four at the surface (decreasing to 0 by 1 km a.g.l).

Previous studies have used cloud base height (CBH) derived from a collocated ceilometer as an additional constraint on the retrieval which allows the retrieval of below-cloud thermodynamic profiles from the AERI in the presence of thick clouds (Turner et al., 2007a; Turner and Löhnert, 2014; Blumberg et al., 2015). Because we focus on radiation fogs under otherwise clear skies, which are, by definition, based at the surface, we initially assumed that the CBH is 5 m a.g.l in all cases. However, large temperature biases (> 5 K) in the AERIoe retrievals during fog illustrated that the TROPoe is highly sensitive to the CBH assumption when using AERI data as an input (see section 3.1). For the final retrievals, we used CBH from the ceilometer to constrain the retrieval if the ceilometer detects a cloud base within 10 minutes of the retrieval time, or if the ceilometer reports obscured vertical visibility, then the detected vertical visibility height (Morris, 2016) is input as the CBH. If neither of these situations occurs, the CBH assumption defaults to 5 m a.g.l. The sensitivity of the retrievals to this choice is discussed further in section 3.1.

The retrieval algorithm outputs $1\sigma$ uncertainties for all variables that incorporate the random error from the observations, the correlated error propagated from the prior, and the sensitivity of the forward model. Errors related to the CBH assumption or phase assumption (when only liquid water is considered) are not included, and we discuss these below in section 3.1 and section 4.

## 2.4 Evaluation metrics

To evaluate the two TROPoe retrieval configurations (AERIoe and MWRoe) we focus on three aspects that are crucial for making accurate fog forecasts with NWP models, for visibility nowcasting, and for understanding the climatic impact of fog:

1. Accurate representation of the structure of the temperature and humidity profile in the lowest 1 km a.g.l.

2. Detection of the presence and strength of shallow surface-based temperature inversions that typically portend the formation of radiation fog.



3. Detection of the initial increase in LWP that signifies the onset of fog and a reduction in horizontal visibility.

To evaluate the accuracy of the temperature and humidity profile retrievals in the lowest 1 km a.g.l, we assess the performance of the MWRoe and AERIoe against 16 coincident radiosonde profiles by evaluating the mean bias and spread between the radiosonde profiles (truth) and the retrievals. We use modified Taylor diagrams (Taylor, 2001; Turner and Löhnert, 2014) to
assess how well the retrieved profiles capture the shape of the true profiles by considering the Pearson's correlation coefficient and the ratio of the standard deviation of the retrieval to that of the truth profile. These results allow for a direct comparison with Blumberg et al. (2015) who compare MWRoe and AERIoe retrievals against a larger number of radiosonde profiles (127) in southwestern Germany (but only consider clear sky days or clouds with bases > 500 m a.g.l).

To evaluate the ability of each retrieval to detect the occurrence and strength of surface-based temperature inversions, we
compare the retrieved surface (0 m) and 10 m temperatures with measurements from in-situ temperature sensors (see section 2.1). We define the inversion 'strength' as the 10 m – 0 m temperature and evaluate the Pearson's correlation coefficients and root-mean-squared error (RMSE) between the retrieved values and the 'truth' (the in-situ temperature sensors). Klein et al. (2015) performed a similar analysis comparing the temperature difference between 10 and 100 m a.g.l. Although we are limited by a maximum sensor height on the tower, we expect the 10 m - 0 m temperature difference to be a good indicator of whether
the retrieval captures the surface-based temperature inversion since most of the inversion during radiation fog often occurs in the lowest 10 m a.g.l (Price, 2011; Izett et al., 2019).

Finally, in the absence of an independent method of determining LWP, we evaluate the ability of each retrieval to detect the initial increase in LWP associated with fog formation and visibility reduction by defining 'fog onset' as where the retrieved LWP minus $2\sigma$ uncertainty (which is directly computed by TROPoe) increases above 0.1 g m$^{-2}$ for at least 10 minutes,
and then compare the difference in fog onset detection time between the MWRoe and the AERIoe for each case study. This methodology allows us to identify when each retrieval identifies a LWP that is above the measurement noise level, hence comparing the sensitivity of the two sets of retrievals to small increases in liquid water that can begin to reduce visibility and impact radiative energy fluxes at the surface.

## 3 Results

### 3.1 Retrieval performance and sensitivity to cloud base height assumption

All 2334 retrievals from both AERIoe and MWRoe converged, meaning that the retrieval algorithm was able to find a solution within the maximum number of iterations. The mean root mean squared error (RMSE) between the final forward model calculation and the observed PAERI radiance across all AERIoe retrievals was 0.54 ±0.09 RU, which is of the order of the instrument noise level (Blumberg et al., 2015). For the MWRoe retrievals the mean RMSE between the final forward model
calculation and the MWR brightness temperatures was 0.38±0.12 K, again within the instrument noise (Rose et al., 2005).

The AERIoe retrievals were very sensitive to the cloud base height (CBH) assumption. Because we are only considering cases of radiation fog under otherwise clear skies, the first iteration of retrievals assumed that, if liquid water was detected, the





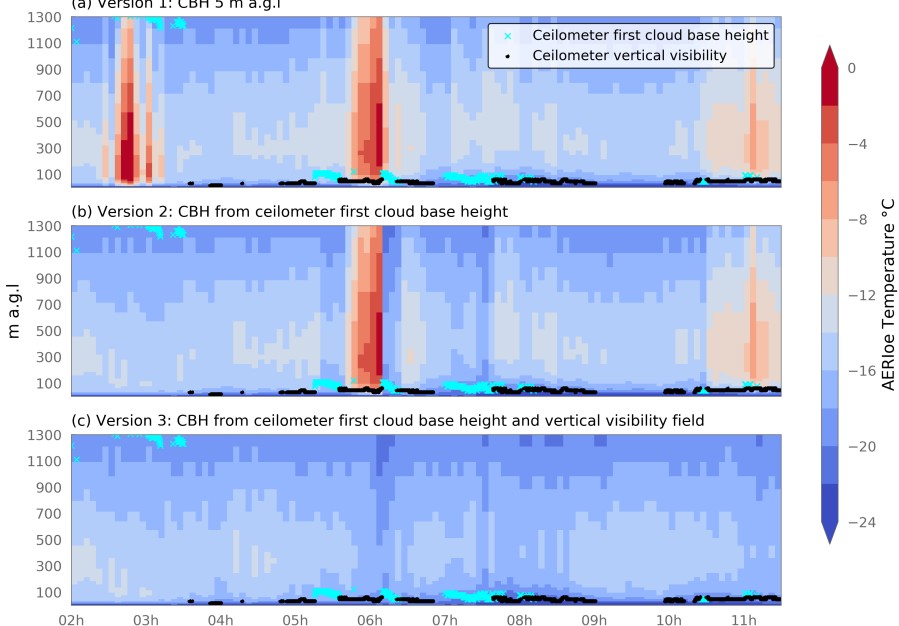

**Figure 3.** Temperature profiles during the 05 Sept case study from three iterations of the AERIoe retrieval using different CBH assumptions. The ceilometer first cloud base height field is overlaid as cyan crosses and the vertical visibility field is overlaid as black circles. (a) Assumed CBH was 5 m a.g.l any time LWP > 0, (b) assumed CBH was set to the ceilometer 'first cloud base height' if the ceilometer detected a cloud within the 10-mins centred on the retrieval time, otherwise 5 m a.g.l, (c) as in (b) except if the ceilometer reported 'obscured' the CBH was set equal to the ceilometer maximum vertical visibility field.

cloud base height was 5 m a.g.l, removing the requirement for additional instrumentation to detect CBH. Figure 3a demonstrates that under this assumption, the AERIoe retrieved unrealistic temperature profiles in some cases when the fog was optically thick
(i.e. ceilometer is obscured), or when the ceilometer detected a cloud with a base close to 1,300 m a.g.l prior to the start of the fog event. The unrealistic temperature profiles manifest as exceptionally warm temperatures just above the surface, indicated by red colours in Fig. 3.

Previous AERIoe algorithms have used the ceilometer 'first cloud base height' field (an output of the Vaisala proprietary software) to estimate CBH (Turner and Löhnert, 2014; Blumberg et al., 2015). Using this assumption rather than simply assum-
ing the CBH to be 5 m a.g.l reduced the temperature profile artifacts in some, but not all cases (fig. 3b). The ceilometer software also outputs a 'vertical visibility' field (described in Morris, 2016) when the extinction profile is such that the atmosphere is obscured but no distinct cloud base can be determined, as often happens in the case of thick fog. The remaining artifacts in the example case study occur when the ceilometer reports a vertical visibility value (Fig. 3b). When we used the vertical visibility field in addition to the 'first cloud base height' field to provide the CBH assumption in the AERIoe retrieval, the remaining
artifacts were removed (Fig. 3c).



Even if the fog is reducing visibility at the surface, the 'cloud base height' is the height at which the fog becomes optically thick to the PAERI, which is similar to when it becomes optically thick for the ceilometer (which uses radiation in the near-infrared, 905 nm). Figure 3 shows that a difference in CBH of just 30 m can make a significant difference to the retrieval when the cloud is close to the surface, demonstrating that accurate CBH measurements are a necessary input to the AERIoe

for retrieving thermodynamic profiles in the presence of fog or near-surface clouds. In contrast, the MWRoe was not sensitive to the CBH assumption, because clouds are markedly more transparent and microwave frequencies. For the remainder of this study, all retrievals (AERIoe and MWRoe) derive CBH from both the 'first cloud base height' and 'vertical visibility' field of the ceilometer.

### 3.2   Performance of retrieved thermodynamic profiles in the lowest 1 km a.g.l

The AERIoe temperature profiles (0 - 1 km a.g.l) compared extremely well to the 16 radiosonde profiles considered, with a mean bias of -0.45°C and a mean RMSE of 1.0°C (Fig. 4a). The MWRoe temperature profiles exhibited a vertically consistent negative bias compared to the radiosonde profiles with an average value of -1.5°C and an average RMSE of 1.6°C (Fig. 4a). Although investigating the source of the negative bias in the MWRoe temperature profile is outside the scope of this study, such systematic biases can often be corrected for, and the similar spread in bias magnitude between the AERIoe and the

MWRoe temperature retrievals imply that the performance of the two sets of retrievals would be similar after an additional bias correction (Fig. 4a). For both sets of retrievals, the temperature RMSE is largest in the lowest 50 m a.g.l where it approaches 2.0°C (Fig. 4a).

For water vapor, the performance of the AERIoe and MWRoe retrievals compared to the radiosonde profiles was very similar (Fig. 4b). Neither set of retrievals exhibited a mean bias, and the RMSE of the AERIoe retrievals was slightly smaller than the

MWRoe up to 800 m a.g.l with a mean value of 0.38 g kg$^{-1}$ (compared to 0.43 g kg$^{-1}$ for the MWRoe).

Figures 4c and 4d show that both sets of temperature profile retrievals are better correlated with the radiosonde profiles (r > 0.98, Fig. 4c,) than the water vapor profile retrievals, which in some cases have correlation coefficients < 0.6 (Fig. 4d), but the spread in the standard deviation ratio is similar (0.9 - 1.2). The AERIoe retrievals (for both temperature and water vapor) have a similar spread in correlation coefficients and standard deviation ratios to the MWRoe retrievals, implying that the performance

of the two retrievals was comparable across this subset of profiles.

### 3.3   Characterisation of shallow surface-based inversions

The AERIoe temperature retrievals at 0 and 10 m a.g.l are equally well correlated with observations (r = 0.99, RMSE = 1.1°C), implying that the AERIoe captures vertical temperature gradients in the lowest 10 m of the atmosphere well and retrieves surface temperatures consistently with high accuracy (Fig. 5). In comparison, the MWRoe retrievals perform worse than the

AERIoe at both heights, with r = 0.86, RMSE = 4.1°C at 0 m, and r = 0.95, RMSE = 2.1°C at 10 m (fig. 5). Notably, the performance of the MWRoe is worst at 0 m, where the MWRoe typically has a warm bias at colder temperatures (72% of the time when T < -7°C) and a cold bias at warmer temperatures (93% of the time when T > -7°C). This bias reduced at 10 m, implying that the temperature lapse rate between 0 and 10 m a.g.l is often incorrect in the MWRoe retrieval (Fig. 5).



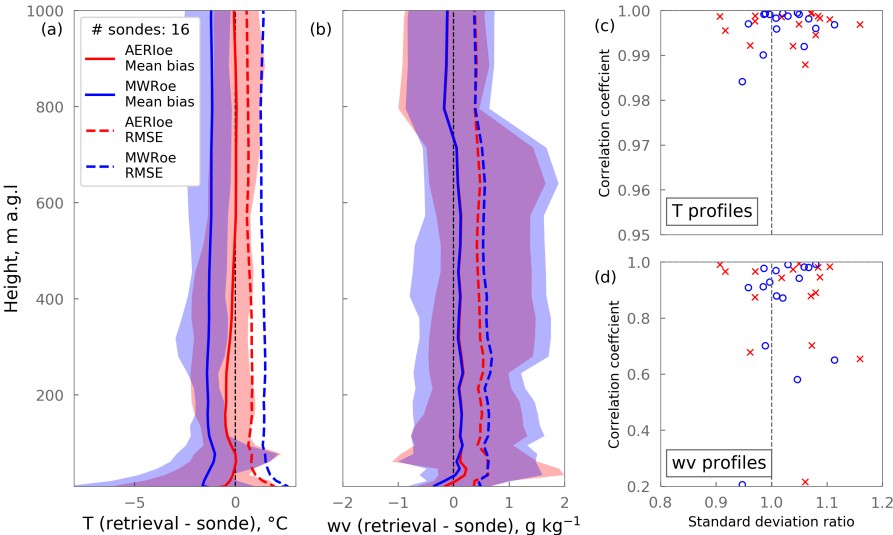

**Figure 4.** Comparison between retrieved thermodynamic profiles and the 16 coincident radiosonde profiles in the lowest 1 km a.g.l. (a) The temperature and (b) and water vapor bias (retrieval – radiosonde) for AERIoe retrievals (red) and MWRoe retrievals (blue), the solid line shows the mean bias and the shaded represents the range. Dashed lines show the root mean squared error (RMSE) (c) The temperature and (d) water vapor modified Taylor plots showing the relationship between the correlation coefficient and standard deviation ratio for each retrieval/radiosonde pair ([1,1] represents a perfect score) for the AERIoe retrievals (red) and the MWRoe retrievals (blue).

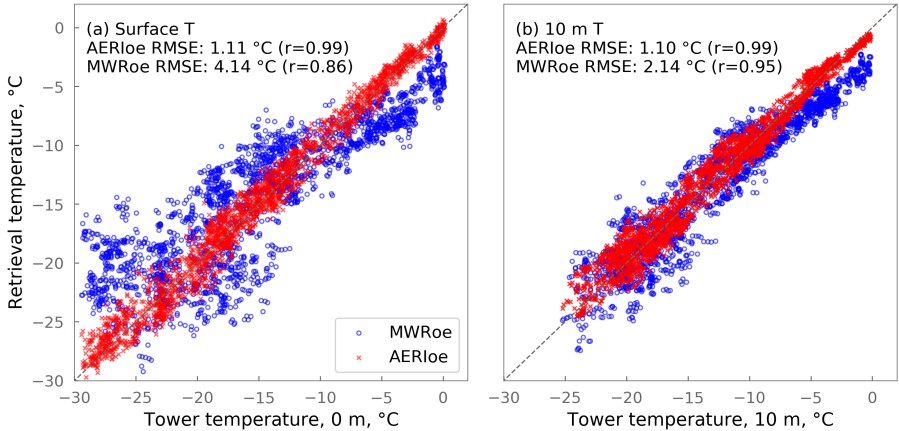

**Figure 5.** MWRoe (blue circles) and AERIoe (red crosses) retrieved temperature versus in-situ temperature measurements from tower mounted HMP155 probes at the surface (a) and 10 m (b). The dashed grey line represents perfect agreement. The Pearson's correlation coefficient (r) and root mean squared error (RMSE) between each set of retrievals and the tower measurements are included on the figure.

Figure 6 shows the poor performance of the MWRoe in retrieving the 0-10 m temperature lapse rate by demonstrating that there is no correlation between the measured surface inversion strength (10 - 0 m T) and that retrieved by the MWRoe. In fact,






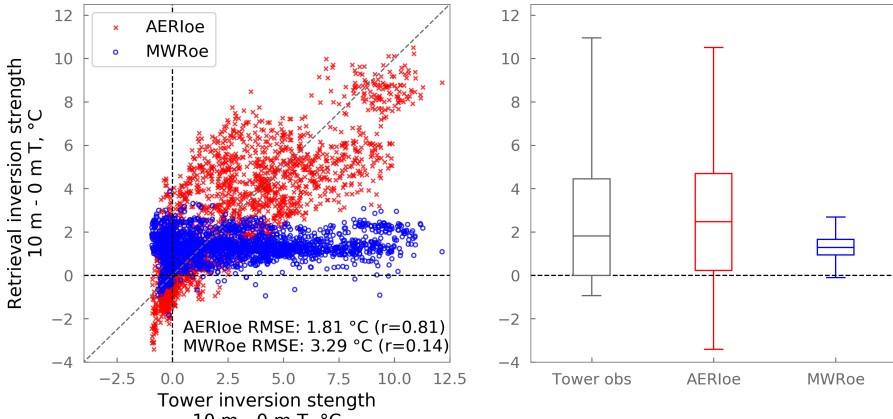

**Figure 6.** (a) Inversion strength $(10\,\text{m} - 0\,\text{m T})$ retrieved from the AERIoe (red) and the MWRoe (blue) versus observations from the tower mounted HMP155 probes. The diagonal grey dashed line represents perfect agreement and the horizontal and vertical dashed black lines at $0°\text{C}$ delineate when a surface based temperature inversion is present (right quadrants) versus absent (left quadrants). (b) The distribution of inversion strength recorded by the tower observations, the AERIoe, and the MWRoe.

in most cases the surface inversion strength retrieved by the MWRoe is $\sim 1°\text{C}$, close to the surface inversion strength in the retrieval prior $(0.7°\text{C})$. This suggests that most of the information about the change in temperature between 0 and 10 m comes from the prior rather than the observations. In contrast, the AERIoe surface inversion strength is well correlated with in-situ measurements $(r=0.81)$ with a RMSE of $1.8°\text{C}$ (Fig. 6). Figure 6 demonstrates that the AERIoe can accurately retrieve shallow

surface temperature inversions with lapse rates of up to $-1.2°\text{C m}^{-1}$.

The reason that the AERIoe can accurately retrieve shallow surface-based temperature inversions but the MWRoe cannot is because the PAERI infrared radiance measurements theoretically contain much more information about the near-surface temperature profile than the MWR brightness temperatures (Blumberg et al., 2015). Figure 7 supports this, by illustrating that the degrees of freedom for signal for temperature from the AERIoe retrievals is greater than that from the MWRoe retrievals,

especially at the surface. The degrees of freedom for signal is a measure of the number of independent pieces of information from the observation vector that the retrieval used to generate the solution (Rodgers, 2000). Figure 7 shows that the AERIoe has around six times as much information about the surface temperature than the MWRoe, and twice as much at 10 m.

### 3.4 LWP retrievals and the detection of fog onset

The LWP retrievals from the AERIoe and MWRoe were well correlated $(r=0.88)$ and fell within $\pm 5\,\text{g m}^{-2}$ of each other in

$\sim 90\%$ of all retrievals, however on occasions there were discrepancies of up to $10\,\text{g m}^{-2}$ (Fig. 8), which can be equivalent to significant differences in horizontal visibility and net surface radiative forcing (see examples in section 1).

Although we do not have an independent measure of the 'true' LWP, Fig. 9 illustrates the difference in sensitivity of the MWRoe and the AERIoe to LWP as a function of LWP magnitude. The PAERI radiance observations are very sensitive to



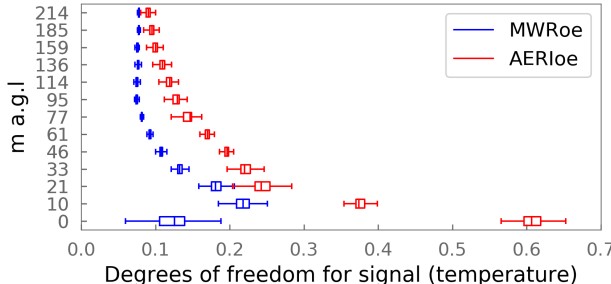

**Figure 7.** Distribution of the degrees of freedom for signal near the surface for all temperature retrievals from the MWRoe (blue) and AERIoe (red).

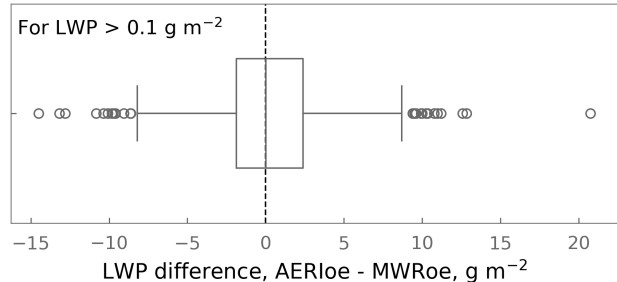

**Figure 8.** The differences in retrieved liquid water path (LWP) between all AERIoe the MWRoe retrievals (LWP > 0.1 g m$^{-2}$). 90% of all retrievals fall within the box plot 'whiskers', the remaining data are plotted as outliers (circles).

changes in LWP when the LWP is small, and so the $1\sigma$ uncertainty in the retrieved LWP from the AERIoe is less than 1 g

m$^{-2}$ for LWP < 20 g m$^{-2}$ (or less than 10%, Fig. 9). In contrast, the uncertainties in LWP derived from MWR brightness temperatures are related to absolute radiometric uncertainties that are approximately constant with respect to LWP, equating to at least 50% uncertainty for LWP < 10 g m$^{-2}$ (Fig. 9). However, as the LWP approaches opacity in the infrared (> 40 g m$^{-2}$) the sensitivity of the PAERI radiance observations to changes in LWP decreases until the uncertainties in the LWP retrievals from the AERIoe become equivalent to those from the MWRoe ($\sim$ 3 g m$^{-2}$ or $\sim$ 6% uncertainty at 50 g m$^{-2}$).

The high sensitivity of the AERIoe to changes in LWP when LWP is small means that the increase in LWP associated with the development of radiation fog under clear skies is detected earlier in the AERIoe retrievals compared to the MWRoe retrievals. The increased sensitivity of the infrared over the microwave to small LWP values was described in (Turner, 2007b). This is illustrated in Fig. 10, which shows the development of LWP during the 15 July 2019 case study. The AERIoe LWP minus $2\sigma$ uncertainty increases to above 0.1 g m$^{-2}$ shortly before 23 h on 15 July and continues to increase gradually, at a

rate of $\sim$ 1.4 g m$^{-2}$ h$^{-1}$, until 02 h on 16 July, after which it increases rapidly to $\sim$ 30 g m$^{-2}$ at 03 h (Fig. 10). In contrast, due to the larger uncertainties in the MWRoe LWP retrieval, the MWRoe LWP does not increase to a value that is significantly different from the noise for at least 10 minutes until the onset of the rapid LWP increase just after 02 h. Despite the relatively





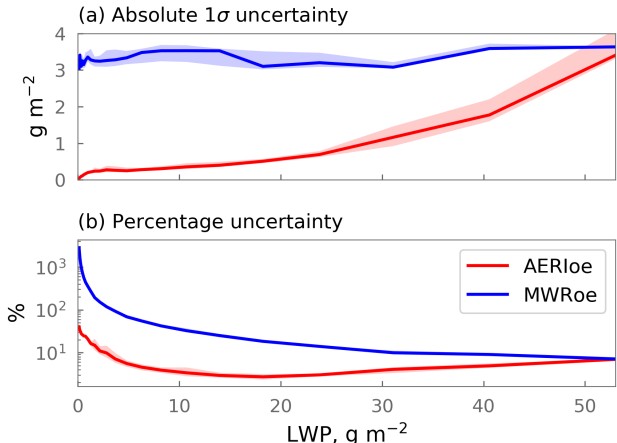

**Figure 9.** $1\sigma$ uncertainty in the AERIoe (red) and MWRoe (blue) liquid water path (LWP) retrievals as a function of LWP. (a) Shows the absolute uncertainties and (b) shows the percentage uncertainties. The solid line is the median of all retrievals and shading is the interquartile range.

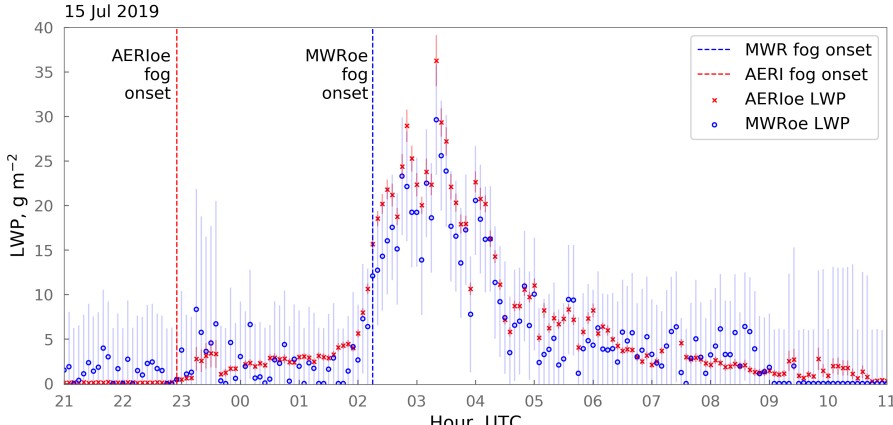

**Figure 10.** The evolution of fog liquid water path (LWP) during the 15 July 2019 case study retrieved from the AERIoe (red) and the MWRoe (blue). Error bars show the $2\sigma$ uncertainty of the retrievals. The vertical red line shows the fog onset determined from the AERIoe retrievals and the vertical blue line shows the fog onset determined by the MWRoe retrievals 3 hours later.

low LWP, the visibility at 00h was only 400 m and the observer reported freezing fog. In this case if the MWRoe LWP retrieval was used to detect fog or for visibility nowcasting, the fog at 00h on 16 July would not have been detected, whereas if the AERIoe LWP retrieval was used instead, it would have been.

The AERIoe retrieval consistently detects the onset of fog (via the increase in LWP) before the MWRoe retrieval (Fig. 11). Only for the 03 Aug case study does the MWRoe detect fog onset 30 minutes before the AERIoe; for all other cases the



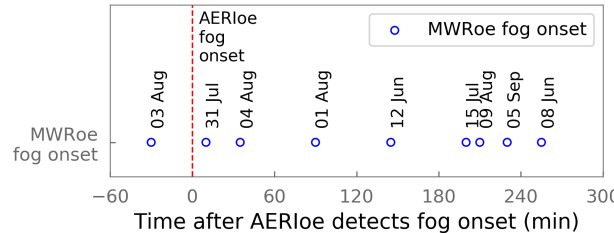

**Figure 11.** Time between fog onset detection from the AERIoe (vertical red line, t=0) and fog onset detection from the MWRoe for each fog case study where both sets of retrievals detected fog (blue circles). Note that for the 4 retrievals not included (13 July, 05 Aug, 14 Aug and 30 Sept), the MWRoe does not detect fog at all during the event whilst the AERIoe does.

AERIoe detects the fog before the MWRoe by up to 255 minutes (Fig. 11). In the four cases that are not shown on Fig. 11 (13 July, 05 Aug, 14 Aug and 30 Sept), the MWRoe never detects the fog whilst the AERIoe does. For all these cases where the MWRoe does not detect the fog, the mean LWP detected by the AERIoe is very low ($< 2.5$ g m$^{-2}$).

## 4 Discussion

Central Greenland provides an excellent opportunity to study climatologically relevant radiation fog due to the pristine environment, commonality of events, and presence of the ICECAPS long term multi-instrument platform. Nonetheless it is a unique environment and therefore the applicability of the results of this study in other environments is not guaranteed. Comparing retrievals between locations is complicated by the dependence on the quality of different prior datasets and instrument calibrations. However, in general, the performance of the MWRoe and AERIoe thermodynamic profile retrievals in the lowest 1 km a.g.l are comparable to the performance assessed in a similar way against 127 radiosonde profiles in southwestern Germany (Blumberg et al., 2015). Blumberg et al. (2015) found the mean RMSE in the temperature profiles (lowest 1 km a.g.l) to be $\sim$ 0.9°C for the AERIoe and $\sim$ 1.1°C for the MWRoe (compared to 1.0°C and 1.6°C in this study), and for water vapor profiles an RMSE of 0.7 g kg$^{-1}$ for the AERIoe and 1.0 g kg$^{-1}$ for the MWRoe (compared to 0.38 g kg$^{-1}$ and 0.43 g kg$^{-1}$ in this study). Nevertheless, under certain conditions, we might expect the performance of the AERIoe to deteriorate. For example, in environments where the total column water vapor is very high (e.g. tropical regions), the atmosphere will have a higher opacity in the infrared and the AERI sensitivity will be reduced (Löhnert et al., 2009). Additionally, the high sensitivity of the AERI, which makes it so suited to the study of fog, also makes it sensitive to localised plumes of pollution or smoke and in some cases to atmospheric aerosols (e.g. Turner and Eloranta, 2008). Both sets of retrievals are also sensitive to the quality of the prior that is used to constrain the retrieval and provide a first guess (Turner and Löhnert, 2014), and to the calibration and characterisation of the particular instrument- which is typically more challenging for the MWR (see Appendix A, Blumberg et al., 2015, and Löhnert and Maier, 2012). All of these factors may impact performance at different locations and should be considered during experimental design. Note that neither instrument operates effectively in rain.





The radiation fog case studies presented in this study are all composed of supercooled water droplets, with some occurring at surface temperatures as low as -27°C (table 3). Supercooled fogs at such cold temperatures are common at Summit, whereas ice fogs during the summer are rare (Cox et al., 2019), and observations of 'fog bows'- atmospheric optics associated with the scattering of light by liquid water droplets- during most case studies confirm the presence of liquid water. However, the possibility exists that some (or even all) of these case studies contain ice crystals in addition to liquid water droplets. The

scattering and absorption properties of ice crystals can be quite different to those of water droplets at wavelengths that are relevant for the AERIoe (e.g. Turner, 2005, Rowe et al., 2013), potentially resulting in biases in the AERIoe LWP retrievals that assume a liquid only cloud. The lack of an independent 'truth' value for LWP means that we cannot quantify any such biases. Nevertheless, the smaller uncertainties in the AERIoe LWP retrieval relative to the MWRoe LWP retrieval are related to the physical sensitivity of the measurement, and so we can expect this result to be consistent across other cases of warm fog.

In this study, we do not use in-situ measurements of surface temperature and water vapor to constrain TROPoe, allowing us to use these measurements as an independent 'truth' against which to evaluate the retrievals. However, it is possible to include surface measurements in the TROPoe observation vector, and for operational purposes surface in-situ measurements are typically used to constrain MWR thermodynamic profile retrievals (e.g. Cimini et al., 2015). . . . Results of Appendix B. . .

Previous studies have demonstrated that a combined instrument approach can provide optimal thermodynamic profile and

LWP retrievals throughout the boundary layer and under all-sky conditions (Turner et al., 2007a; Löhnert et al., 2009; Turner and Lohnert, 2021; Smith et al., 2021; Djalalova et al., 2021). However, for the conditions relevant for the formation of radiation fog (the near surface thermodynamic profile under clear skies or in the presence of optically thin fog) the performance of these combined retrievals is equivalent to the performance of retrievals that only use AERI measurements. For this reason, we do not consider a combined instrument approach in this study, however for multi-purpose applications that require accurate

near-surface retrievals in addition to above-cloud thermodynamic profiles, a combined instrument approach might provide the optimal solution.

## 5    Summary and conclusions

Previous studies have demonstrated that AERI measurements of spectral infrared radiance are more sensitive to the structure of the near-surface temperature profile and to small changes in liquid water path (LWP) than MWR measurements of microwave

brightness temperatures (Turner, 2007b; Löhnert et al., 2009; Blumberg et al., 2015). The purpose of this study was to compare the results using a consistent physical retrieval algorithm (TROPoe) based on observations from the two instrument types during cases of supercooled radiation fog in central Greenland (the AERIoe retrieval based on observations of infrared radiance and the MWRoe based on microwave brightness temperatures). We assess the performance of the two retrievals against three criteria that are critically important for the forecast and detection of radiation fog:

1. Ability to retrieve accurate thermodynamic profiles in the lowest 1 km a.g.l of the atmosphere.

   2. Ability to capture the strength and development of shallow surface-based temperature inversions that typically portend the formation of radiation fog.





3.  Ability to detect the initial increase in LWP that signifies the onset of fog and a reduction in horizontal visibility.

Although there are only 16 coincident radiosonde profiles available for comparison, the bias and RMSE statistics of the
temperature and water vapor profiles in the lowest 1 km a.g.l are consistent with the findings of Löhnert et al. (2009) and
Blumberg et al. (2015), suggesting that the performance of both sets of retrievals in the Arctic and under conditions for the
formation of supercooled fog are similar to the performance in the mid-latitudes and under other sky conditions (clear skies
or below clouds with bases above 500 m a.g.l). We find that the water vapor profile retrievals in the lowest 1 km a.g.l are
comparable for both the MWRoe and the AERIoe with RMSE of 0.43 g kg$^{-1}$ for the MWRoe and 0.38 g kg$^{-1}$ for the
AERIoe. The AERIoe temperature profile retrievals perform better in terms of bias and RMSE than the MWRoe for the 16
cases considered (MWRoe bias: -1.5°C, RMSE: 1.6°C; AERIoe bias: -0.45°C, RMSE: 1.0°C), however, the consistency of
the negative temperature bias in the MWRoe suggests that an additional bias correction may be possible that would result in
comparable performance between the two sets of retrievals.

A unique aspect of this study was the assessment of the ability of the two retrieval types to characterise shallow (0-10 m a.g.l)
surface-based temperature inversions. Despite the similar performance in general of the temperature profile retrievals up to 1 km
a.g.l, the ability of the two retrieval types to characterise the 0-10 m temperature lapse rate was markedly different. The AERIoe
0-10 m temperature differences were well correlated with in-situ observations, capturing surface temperature inversions well
up to a lapse rate of -1.2°C m$^{-1}$ (previous studies have demonstrated the ability of the AERIoe to characterise near-surface
lapse rates well for values of -0.01 to 0.01°C m$^{-1}$, Klein et al., 2015). However, the MWRoe 0-10 m temperature differences
were not correlated with observations and did not deviate more than 1°C from the prior. The reason for this difference is that
the infrared radiance measurements from the AERI contain more information about the temperature near the surface than the
MWR measurements.

In addition to increased sensitivity to shallow surface temperature inversions, the AERI is much more sensitive to small
changes in LWP, with the result that the uncertainties in retrieved LWP from the AERIoe are much smaller than those retrieved
from the MWRoe for LWP < 50 g m$^{-2}$. This means that the AERIoe is consistently able to detect small changes in LWP
that signify the onset of radiation fog and reduction in horizontal visibility by up to 255 minutes before the MWRoe. This has
important implications for fog detection and visibility nowcasting, because even a very small LWP (< 5 g m$^{-2}$) can reduce
horizontal visibility, and the MWRoe alone would not have detected fog on some occasions when reported visibility was as
low as 400 m.

Based on these results, we hypothesise that the assimilation of near-surface temperature profile retrievals from an AERI
into NWP models could improve fog forecasts beyond the improvements already seen through the assimilation of MWR
measurements (Martinet et al., 2020). In addition, the increased sensitivity of the AERI to small changes in LWP (compared to
the MWR) will allow the AERI to detect fog onset earlier and miss fewer fog events, with the potential to increase the skill of
fog nowcasting products and improve climatological analyses of fog radiative effects. Although this study demonstrates that
the AERI is particulary well suited to retrieving boundary layer properties that are key for radiation fog formation, there are
trade-offs that must be considered when selecting instruments for operational use; notably that the AERI is unable to retrieve
thermodynamic profiles above optically thick clouds (LWP > 40 g m$^{-2}$), and that the AERIoe retrieval is particularly sensitive





to the CBH assumption during fog / low cloud. The results of this study present a case for future observing system experiments (or observing system simulation experiments, for example as in Otkin et al., 2011; Hartung et al., 2011) to quantify the impact
of the operational use of AERI observations in terms of improvements to NWP skill, particularly in the case of radiation fog.

*Data availability.* ICECAPS data are available from the Arctic Data Center: HATPRO MWR (doi:10.18739/A2TX3568P), MMCR (doi:10.18739/A2Q52FD4V), POSS (doi:10.18739/A2GQ6R30G), and radiosonde profiles (doi:10.18739/A20P0WR53). PAERI data and retrieval output from the TROPoe are in the process of being submitted to the Arctic Data Center and are available upon request. ICECAPS-ACE HMP155 temperature/ humidity sensor data can be accessed through the CEDA archive at http://catalogue.ceda.ac.uk/uuid/
f06c6aa727404ca788ee3dd0515ea61a.

## Appendix A:  MWR Tb bias correction

An external liquid nitrogen target is used to determine the effective temperature of the MWR internal noise diode, which is required to convert the observed signal into brightness temperature (Tb) values as described in section 2.1.2. Due to the personnel and resource requirements, this calibration is only performed twice a year at Summit. Imperfect calibrations can
result in a radiometric bias in the Tb measurements, and drift in the effective temperature of the internal noise diode can occur in between calibrations (Löhnert and Maier, 2012; Blumberg et al., 2015).

To determine this radiometric bias in the observed Tb values, the twice-daily radiosondes launched at Summit between 01 June and 31 August 2019 were used as input in the MonoRTM, and the bias between the observed and computed Tb values was determined. The evolution of the bias over time is well-illustrated by showing the bias values for the cases used in this analysis
(Fig A1). The bias in the K-band channels remained very small (absolute value less than 0.5 K), confirming that the automated tip curve calibration method applied to those transparent channels was working well. However, there is a significant negative bias (calculation larger than the observed radiance) in the 51 to 54 GHz channels, and this bias changes with time. However, the bias in the 51.26 and 52.28 GHz channels is variable with time with no apparent pattern (inset in Fig A1). Fortunately, the Tb bias in the 55 to 58 GHz channels is stable with time, and the magnitude is relatively small (less than 0.7 K for those
channels).

Note that we do not consider or correct for possible spectral biases in the MWR frequencies channels (Löhnert and Maier, 2012). It is possible that the large Tb biases in the 51.26, 52.28, and 53.86 GHz channels are a combination of both spectral and radiometric biases. It is important to note that the biases in those channels might not be entirely due to the calibration accuracy of the microwave radiometer; the bias could also be explained by systematic errors in the radiative transfer model used, as
the various uncertainties of the absorption line properties at those frequencies results in large model uncertainty (Cimini et al., 2018).

Figure A2 illustrates the difference between the MWRoe-retrieved thermodynamic profile compared to the radiosondes during the radiation fog case studies with and without the mean radiosonde-derived bias correction applied. The application of the bias correction reduces the mean bias in the water vapor profile from 0.14 g kg$^{-1}$ to 0.01 g kg$^{-1}$ and the mean RMSE





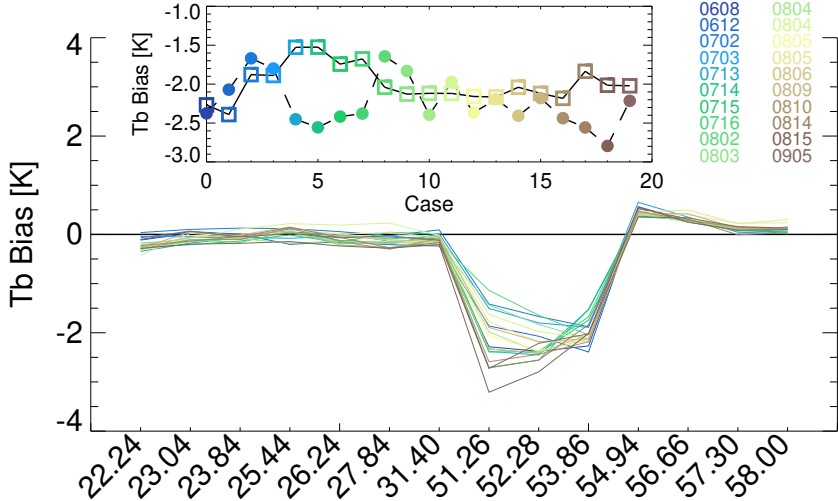

**Figure A1.** The Tb bias offset for the cases used in this analysis. The inset plot shows the temporal variability of the 51.26 (filled circles) and 52.28 (open squares) GHz channels for these cases, where the colors indicate the date (MMDD) of the case.

from 0.47 g kg$^{-1}$ to 0.43 g kg$^{-1}$. However, the bias correction has little effect on the temperature profiles. The reason for the consistent small negative temperature bias in the MWRoe both with (mean -1.45 °C) and without (mean -1.26 °C) the bias correction is currently unknown, especially since the mean Tb bias in the high-frequency end of the V-band is very close to zero.

The reduction in the bias and RMSE of the MWRoe-retrieved water vapor profile compared to the radiosondes demonstrates
that the application of the additional Tb bias correction is essential for making accurate MWR thermodynamic profile retrievals. Currently the only way of performing this bias correction is by using an alternative 'truth' profile (in this case radiosonde profiles); this is a significant disadvantage of the MWR since radiosondes are spatially sparse, resource intensive, and expensive. However, a method using only the climatology data, as encapsulated in the a priori data, has been proposed by Djalalova et al. (2021); this new method helps to account for some of the spectral artifacts in the bias but needs additional research to help
characterize any systematic error that might be introduced by the method.

## Appendix B: The impact of constraining the retrievals by surface meteorological data.

Operational retrievals of thermodynamic profiles from MWRs are typically constrained by an in-situ measurement of surface temperature, usually from a sensor that is integrated with the MWR (e.g. Cimini et al., 2015). We did not include this constraint in the TROPoe retrievals (either AERI- or MWR-based) to allow for the use of the in-situ surface temperature as an indepen-
dent validation measurement. To illustrate the impact of including the in-situ surface temperature in the MWR retrieval, we ran another set of retrievals (MWRoe-sfc) which is identical to the MWRoe except that it includes the in-situ surface temperature





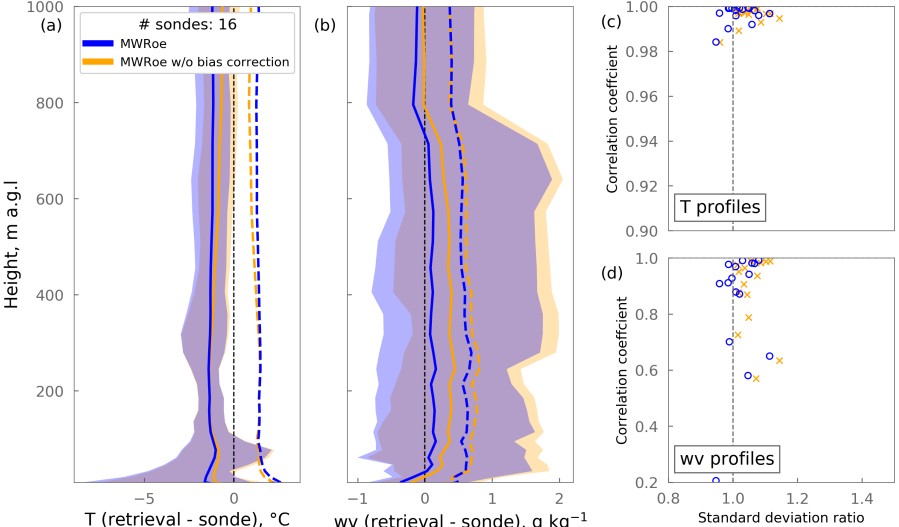

**Figure A2.** As figure 4 but comparing the performance of the final MWRoe retrieval (blue) with that of the MWRoe retrieval without the additional Tb bias correction applied (orange). The solid line shows the mean bias and the shaded represents the range. Dashed lines show the root mean squared error (RMSE).

and water vapour observation from the HMP155 sensor in the observation vector. Since the microwave brightness temperatures contain very little information about the surface temperature (Fig. 7), the additional constraint dominates the surface temperature retrieval so that the 'retrieved' surface temperature from the MWRoe-sfc is essentially the same as the in-situ temperature

measurement (Fig. B1a). By aligning the MWRoe-sfc surface temperature with the in-situ measurement, the representation of the surface temperature inversion defined as the difference between the 10 m and surface temperature is naturally improved. However, note that the 10 m temperatures retrieved by the MWRoe-sfc perform only marginally better than that of the MWRoe compared to the 10 m in-situ measurement, and not as well as the AERIoe which did not include the extra information from the in-situ surface temperature probe (Fig. B1b). This implies that constraining the retrieval by the in-situ surface temperature

does not translate to improvements in the temperature profile retrieval above that level.

Figure B2 shows that the mean degrees of freedom for signal for the surface temperature retrieval is over four times higher for the MWRoe-sfc compared to the MWRoe due to the additional information about the surface temperature in the observation vector. However, above the surface, the AERIoe still contains more information about the temperature profile than the MWRoe-sfc in the lowest 500 m of the boundary layer; the region in which accurate temperature profiles in NWP models are critical for

successful fog forecasts (Martinet et al., 2020).





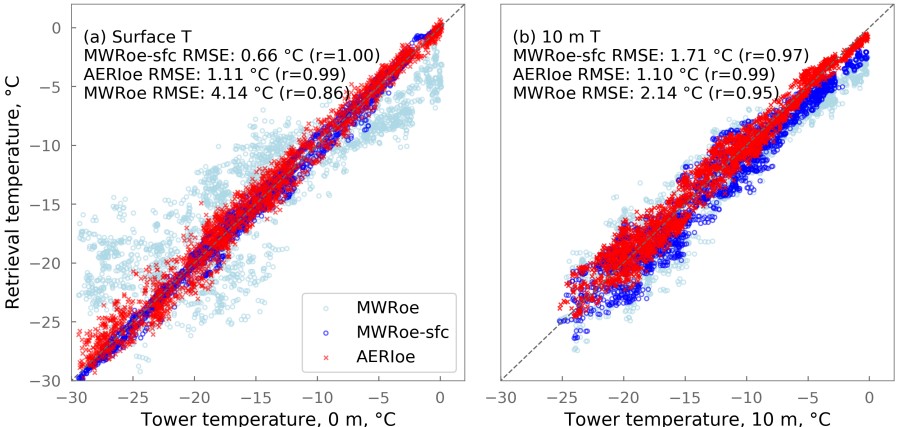

**Figure B1.** Retrieved temperature versus in-situ measurements from the tower at the surface (a) and 10 m (b). The MWRoe retrievals used in the main study are plotted in pale blue, the AERIoe retrievals in red, and the MWRoe retrievals that are constrained by the surface temperature (MWRoe-sfc) are in blue.

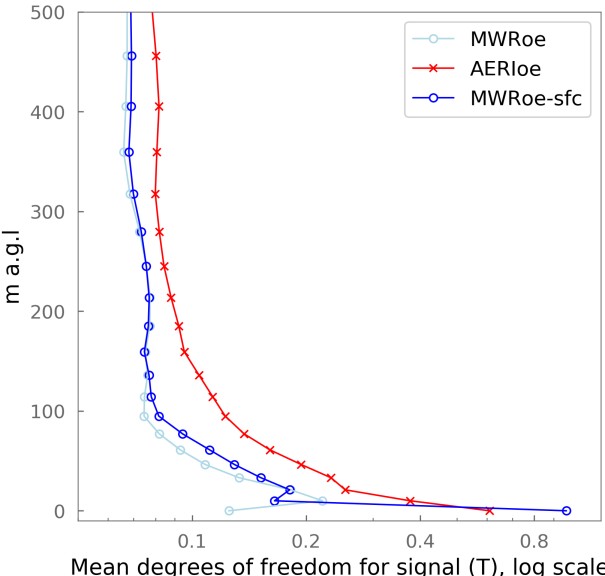

**Figure B2.** Mean degrees of freedom for signal across all temperature retrievals from the MWRoe (pale blue), the AERIoe (red), and the MWRoe constrained by the in-situ surface temperature measurement, MWRoe-sfc (blue).

*Author contributions.* DT conceived the study, developed and ran the TROPoe retrieval algorithm, processed the MWR data, calculated the Tb bias correction, and contributed to the text. VW prepared the PAERI data and contributed to the text. HG led the analysis with contributions from DT and RN. HG prepared the manuscript with contributions from all co-authors.



*Competing interests.*  The authors declare that they have no conflict of interest.

*Acknowledgements.*  The efforts of technicians at Summit Station and science support provided by Polar Field Services were crucial to maintaining data quality and continuity at Summit. ICECAPS is a long-term research program with many collaborators, and we are grateful for all their efforts in developing and maintaining the various instruments and data products used in this study. Financial support for ICECAPS-ACE was provided by NSFGEO-NERC grant 1801477, and HG was funded by the NERC SPHERES DTP grant number NE/L002574/1. We would like to thank Maria Cadeddu, Argonne National Laboratory, for her work in updating the HATPRO's calibration using automated tip

curves. Finally we are grateful for feedback from the PROBE-COST fog alerts meeting (23 Nov 2021) that improved this manuscript.



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
