# Peer review of "Passive ground-based remote sensing of radiation fog"

_Atmospheric Measurement Techniques, 2022_

## Referee Comment (RC3)

**Review AMT :**
**Passive ground-based remote sensing of radiation fog**

**Overview:**

This paper investigates the capability of two passive instruments, the AERI and HATPRO MWR to detect shallow surface-based temperature inversions and to provide good retrievals of LWP under thin radiative fog events. I think the paper is very well written and of a very good scientific quality. It is also very interesting and important for the scientific community as this is the first time the benefit of the AERI instrument for fog forecast is evaluated. However, there are a few points that I think are important to clarify before the publication of this manuscript. I would recommend a more moderated conclusion through the manuscript instead of trying to prove that AERI is better than the MWR for fog forecast improvements. In fact, I think the major issue of the lack of visibility data should really limit the conclusions that AERI can really detect fog onset before the MWR. To be more objective, with the current dataset used in the paper, what is demonstrated is an increase signal in LWP detected earlier with AERI compared to the MWR but without « real » proof that this is related to fog presence. Secondly, I think the discussion on the temperature lapse rate should also be more widely discussed in the paper because in-situ temperature surface from the MWR should be integrated in the 0-10m lapse rate comparisons. I also believe that the temperature inversion over a thicker layer might already be a good proxy for radiative fog and the comparison considering only a 10m thick layer is clearly penalizing the MWR.

Finally, I think a discussion about the use of LWP retrievals for fog nowcasting dissipation (Toledo et al 2021, ACP) would be beneficial to the article. In fact, even if AERI might detect fog formation a bit earlier due to very low LWP values, what about potential limitations for fog dissipation when the fog is thicker and might reach the saturation signal of the AERI ? Would the AERI be a good candidate to apply the conceptual model described in Toledo et al 2021 or the MWR would be a better candidate this time ?

This question might probably lead to a more balanced conclusion not trying to put the AERI against the MWR but to open the perspective of the instrumental synergy highlighting the benefit of each instrument depending on the fog stages.

**Major points :**

**Lack of visibility data :**

One of the major issues for me is the lack of visibility measurements which is the « reference » instrument to detect fog events. This is for me especially problematic for figures 10 and 11. I think, lacking this important reference measurements, the authors are going a bit too fast concluding that AERI is able to detect fog onset 4 hours before the MWR. What is true is that we detect a signal of LWP increase in the AERI earlier than observed in the MWR. However, as there is no reference instrument to give the exact time of fog formation, we cannot rely on the fact that an increase of 0.1 g/m² in the AERI LWP determines the true time of fog formation. I think it is particularly important as, if I understood well, the AERI LWP increase could also be due to the presence of ice crystals in the atmosphere that can't be detected by the MWR for example.

To help clarifying this point, can the authors provide with figure 10, the time serie of ceilometer CBH / vertical visibility for that specific day and the time serie of relative humidity observed at the ground and 10m on the tower ? In fact, I would expect even thin radiative fogs to be detected by the ceilometer which should provide a CBH lower than 50m within these conditions or a vertical visibility and could be a first good validation of when the fog really forms. You could also look at the time series of relative humidity (RH) from the tower measurements : though a RH > 95 % is not always a good proxy of the presence of fog, a RH < 95 % would easily show that there is little chance of fog.

I think this lack of visibility data might also be problematic for the definition of the 13 fog cases provided in table 3. I understand that only two criteria have been used : an increase in the downwelling infrared radiances representative of a cloud presence and no cloud detected above 200m by the MMCR. However, as mentioned later in the manuscrit, the ceilometer CBH should be used to detect clouds with CBH below 50m or ceilometer data providing a vertical visibility in general informing of fog presence to at least avoid wrongly classifying low clouds (with a CBH < 200 m) in fog if they do not reach the ground.

It would be interesting to specify for each of the 13 fog events identified the maximal CBH or vertical visibility height within each fog event detected by the CT25K to be sure that no potentially « low louds » have been wrongly classified as fog.

The current methodology used in the paper is particularly questioning when looking at figure 3 which shows the ceilometer CBH for an identified radiative fog starting at 2 UTC and ending at 12 UTC. Here the ceilometer CBH is around 1300 m until 3h30 while table 3 specifies a starting fog time at 2 UTC : could you explain how fog could form even in the presence of this low cloud and why the MMCR does not detect any cloud at that altitude ?

The definition of the fog events used in this study is also questionable when it is mentioned line 223 that, for some fog events, no CBH or vertical visibility is provided by the ceilometer : in that case how can we be sure that the increase in downwelling infrared radiances is not due to ice particles instead of fog presence that would not be detected by the ceilometer ?

**Temperature lapse rate :**
It is well known that MWRs can not provide two independent informations of temperature at surface and 10m a.g.l as their vertical resolution is approximately higher than 50m. This is a common approach to combine the MWR with an in-situ surface station (either the one provided with the HATPRO or an external station as MWRs are often deployed in instrumented sites). This combination is used for atmospheric boundary layer height detection and the detection of stable boundary layer conditions from MWR measurements often based on the temperature difference between a higher altitude level and 50m (and not the MWR retrievals at 0m). In that sense, I think figure 6 is not entirely objective in the way the MWR lapse rate is evaluated and compared against the AERI. However, this figure demonstrates the capability of the AERI to retrieve this lapse rate while the MWR can't without integrating the surface station. For a more balanced conclusion, I would first recommend to include in figure 6, the comparison with the MWR lapse rate calculated with the in-situ surface temperature station integrated with the MWR (as it is shown in Appendix B).
Additionally, I think the authors should also compare the AERI lapse rate and MWR lapse rates together with higher altitude levels (like T100m-T50m). This is important because even if the figure demonstrates the capability of the AERI to better capture the 0-10m temperature inversion, I am not entirely convinced that the 0-10m lapse rate is the « key » proxy of fog formation. I expect temperature inversions during radiative fog to spread over a larger atmospheric layer, where there is a high chance that the MWR can detect the inversion as well as the AERI and could already be an information sufficient to improve fog nowcasting. As for data assimilation, an improved lapse rate between 0 and 10m seems too resolved compared to the capability of current data assimilation systems (and knowing that surface stations are already assimilated in NWP models, I think it is really pertinent to investigate the MWR capability versus AERI on a larger layer than only 10 m…).

**Minor points :**

line 84 : large error during thin fogs → large **relative** errors during thin fogs

line 154 : plus the current and temperature of Stirling cooler : I don't entirely understand the sentence as if one word is missing after « current ».

line 388 : It looks like « Results of Appendix B » should appear within parenthesis and the « ... » before and after should be removed.

---

## Author Comment (AC1)

**AMT-2022-12: Response to reviewers**

03 May 2022

**The co-authors would like to thank all reviewers for their feedback and thoughtful suggestions. We have responded to each comment below. All line numbers refer to lines in the revised document unless otherwise stated.**

**In addition to edits made in response to reviewer comments, we have made the following additional minor corrections which have no further implications on the results:**

1) We have corrected a typo on line 201, the number of verified clear sky hours between June and September 2019 is 179 not 236.
2) We have corrected an error in Fig. 4 panel (d): The original plot showed the temperature standard deviation ratio on the x-axis rather than the water vapor standard deviation ratio.

Original reviewer comments are included in *italics* below. Co-author responses are in simple text, and quotations from the revised manuscript are enclosed in square brackets ["..."].

**Response to RC1:**

*General comments*
* * *
*- Here we only look at cases were we know fog develops. How many false alarms for fog onset to expect? Can you please comment on this?*

Quantifying a false alarm rate depends on the definition of fog, which might be different for different purposes. For example, a horizontal visibility threshold of <1,000 m is a typical definition of fog for transport safety and nowcasting, but from a radiative perspective, much thinner fogs might still have a large impact on the surface energy budget.

We have added the following lines to emphasize this point:

Lines: 184 to 192: ["For forecasting and nowcasting purposes, fog is usually defined by a threshold in horizontal visibility (typically < 1,000 m) which has important implications from a safety perspective (Gultepe et al., 2007). However, limiting the definition of fogs to those that reduce visibility to < 1,000 m encourages thinner fogs (or mists) to be ignored or incorrectly classified as clear sky events. Being able to accurately measure thinner fogs is extremely important because (a) they form the precursor to thick fog, (b) they modify the surface moisture, aerosol, temperature and radiative structure which might impact fog development further down the line (Haeffelin et al., 2013) and (c) they can have important radiative and climatological impacts even without developing into a thick fog (Cox et al., 2019, Hachfeld et al., 2000). Because both the MWR and AERI are directly sensitive to the radiative impact of fog (as opposed to visibility), for the purpose of this study, we define fog as the presence of near surface liquid water that has a detectable radiative impact."]

We have added an additional method of fog detection using the ceilometer for independent verification (see lines 404 to 409 for the description of this). Using the ceilometer as truth, there would be one case out of 12 (case ID 7) where the AERIoe fog detection was a false alarm (i.e. the AERI detected fog when the ceilometer did not). However, even for this case, the onsite observer recorded that a fog bow was present between 07:15 and 08:30, indicating that there were in fact liquid droplets present. This was a marginal case (max LWP 2 g m$^{-2}$) that demonstrates the ability of the AERI to detect miniscule amounts of liquid water when even the ceilometer cannot. Since we do not have access to an instrument that is more sensitive to liquid water than the AERI, it is not possible to determine a false alarm rate for the detection of liquid water presence versus absence.

Of course, for fog nowcasting, the real parameter of interest is the reduction in horizontal visibility. Neither the MWR nor the AERI are suitable for direct measurements of horizontal visibility (which depends on fog depth and droplet size distribution as well as liquid water path). To determine a false alarm rate for fog detection based on a visibility threshold, it would be necessary to develop an algorithm to estimate horizontal visibility from the AERI and MWR measurements, and then use independent measurements of horizontal visibility to determine a false alarm rate for fog detection based on a visibility threshold, this is outside the scope of the present study.

*- The commercial MWR have a surface temperature sensor incorporated. With this cheap addtion, the retrieval can be constrained at the lowermost levels. Excluding these is scientifically possible, however, it does distort reality if we consider the instrument as a whole (MWR plus meteo station). Please account for this in your manuscript, not just in the appendix. You have done all of this in appendix B, and I found it relevant enough to include Fig B1 and B2 to the main manuscript in place of Fig 5 and 7.*

Based on this suggestion, we have integrated appendix B into the main text and replaced figures 5 and 7 with figures B1 and B2. Note that small changes in the RMSE values are due to the removal of one of the case studies, the reasons for which are discussed in the response to Reviewer 3.

*Specific comments*
* * *
*- l. 35:  on clear evenings with light winds -> on clear evenings with no or light winds (otherwise one might think that wind is needed)*

Our preference is to not make this change, it's not possible to measure 'no wind' absolutely which makes the distinction somewhat meaningless. Also, some very light wind (> 0.5 m/s) can aid the formation and development of radiation fog by increasing the depth of the layer that experiences surface-driven cooling (Oke, 2002; Haeffelin et al., 2013).

*- l. 54:  The DIAL by Vaisala claims to have a lowest usable range gate at 50 m (see e.g. Mariani et al. 2021). Anyway, I agree that in the end this does not change a big deal for the usability for fog.*

Thanks for pointing this out, we have changed this to 50 m to reflect the claim by Vaisala. Although I note that none of these papers assess the performance of the Vaisala DIAL below 100 m, this would be nice to see!

*- l. 66: I don't like the +/- here as the direction of the sign of the change has opposite influence on the melting potential.*

Bennartz et al., (2013) show that, for this case, either an increase or a decrease in the LWP would have reduced the surface melt potential. This is because a thicker cloud (higher LWP) would have reduced the amount of shortwave solar radiation reaching the surface, but a thinner cloud (lower LWP) would have reduced the amount of downwelling longwave, with both effects resulting in a decrease in surface melt potential overall.

We have changed this line to the following to avoid confusion here:

Lines 67-71:
["Cloud LWP was a critical control on the exceptional Greenland Ice Sheet melt event of 2012 (Bennartz et al., 2013). At the highest point on the ice sheet, had the cloud LWP been 20 g m-2 higher than observed, the reduction in downwelling shortwave radiation would have prevented surface melt. Equally, had the LWP been 20 g m-2 lower, the reduction in downwelling longwave radiation would have prevented surface melt (Bennartz et al., 2013). "]

*- l. 68: It is an over-simplification to reduce MWR to K- and V-band. What about "... that measure downwelling radiation. Commercial sensors for temperature and water vapour profiling typically operate 14-35 spectral channels at 22-31 GHz and 51-58 GHz.*

We have changed this sentence to the following per your suggestion:

Line 72-73;
["Ground-based microwave radiometers (MWRs) are passive sensors that measure downwelling radiation. Commercial MWRs for temperature and water vapor profiling typically operate 14-35 spectral channels between 22-31 GHz and 51-58 GHz and are sensitive to the lowest 6 km of the atmosphere (Lohnert and Maier 2012; Blumberg et al., 2015)."]

*- l. 77: "Despite the promise of MWRs to improve fog forecasts". Please re-word the beginning of this sentence. I judge that the above-cited references (Martinet, ...) demonstrated that the improvement is more than just a "promise" that might not be fulfilled.*

We have reworded this sentence to the following:

Lines 81-83:
["However, the maximum vertical resolution of boundary layer temperature profile retrievals from the MWR is 50 m at the surface, decreasing to 1.7 km at 1 km a.g.l (Rose et al., 2005, Cadeddu et al., 2013), which is insufficient to resolve the shallow surface-based temperature inversions that often portend the onset of radiation fog (Price et al., 2011, Izett et al., 2019)."]

*- l. 104: your study only assess the situation of supercooled radiation fog. Would you expect your results to also be representative for fog>0°C? Why (not)? Please elaborate in the*

*manuscript. Ok, see you mention it is not guaranteed to be representative in Sect. 4, but maybe you want to give an indication here already.*

We have moved the reference to the discussion of this issue in section 4 up to the end of the introduction section.

Lines 119-120: ["The applicability of the results of this study to other (less extreme) environments, and different types of fog, is discussed in section 4."]

*- l. 173: here you study only cases without cloud. Please comment on what effect clouds above fog would have on AERI observations.*

We are assuming that you mean line 177 in the original manuscript here. Typically, radiation fog forms under clear skies, so it makes sense to use this criterion to identify radiation fog events.
As described in section 3.1, the temperature profile retrievals from the AERI are highly sensitive to the cloud base height assumption, and so an accurate cloud height (i.e. from a ceilometer) would be required to retrieve thermodynamic profiles in the presence of cloud. Also, as pointed out in the introduction, if the total optical depth of the cloud is so thick that the cloud is opaque in the infrared, then the AERI is not sensitive to the atmosphere above the cloud. For the liquid water path retrieval, both instruments would be sensitive to another cloud layer above an optically thin fog, and so we wouldn't be able to separate the fog liquid water path from the cloud liquid water path in this case.

We added a line in section 3.2 to clarify this point:

Lines 192-195: ["Radiation fogs typically form under clear skies and as such we only consider cases of fog under otherwise clear skies, this allows us to be certain that the LWP retrievals are a measure of fog LWP alone."]

*- l. 249: I undertand that you are limited to max altitude 10m by the height of the tower. However, this comparison is most probably artificially penalising MWR which, with its lower vertical resolution, might perform better when comparing to 20 or 50 m. And having a somwhat thicker layer might also make sense from a physics point of view. Please comment on this.*

One of the key points that we are trying to emphasize here is that the low vertical resolution of the MWR means that it is unable to retrieve the strong surface-based temperature inversions associated with radiation fog formation. The surface inversions associated with radiation fog initiate in the lowest 10 m above the surface at Summit, and greatest part of the temperature inversion associated with radiation fogs is often concentrated in this layer (Price 2011; Izett et al., 2019). So, we believe that this comparison is important to emphasize that the resolution of the MWR is insufficient to capture these events. However, w have added an additional panel to Fig. 6 showing the comparison of the 100 m – 10 m inversion strength with the 14 coincident radiosonde profiles, unfortunately the limited number of radiosondes available limit the conclusions we can draw from this. We have added the following text discussing the results of this comparison:

Lines 362 to 369: ["The radiosonde profiles provide an alternative independent measure of surface inversion strength, allowing the comparison of the ability of each retrieval

configuration to capture surface temperature inversions over a deeper layer. Fig 6b compares the 100 m - 10 m retrieved inversion strength with that measured by the 14 coincident radiosonde profiles. Over this depth the RMSE of the AERIoe and the MWRoe-sfc are comparable to the values for the 10 m - 0 m comparison (1.65 and 1.83 C m-1 respectively), but the MWRoe RMSE remains much larger (2.22 C m-1), demonstrating that the MWRoe alone is not capable of accurate retrievals of surface temperature inversions even in this deeper layer. Only the AERIoe retrievals in this case are significantly correlated (r=0.46) with the radiosonde measurements, although the small number of radiosondes available for comparison makes it difficult to draw robust conclusions from this result."]

*- l. 317: "This suggests that most of the information ... comes from the prior" -> It would be interesting to see the measurement response (i.e. the sum of the averaging kernels). This would allow us to judge how much info comes from the prior rather than just guessing.*

Thank you for mentioning this – after examining the averaging kernel I think that my assumption that most of the information comes from the prior is actually incorrect. The reason for the constant difference is that the averaging kernel for perturbations in temperature at '0 m' and '10 m' is almost identical for the MWRoe – it simply cannot see a difference between the two levels, hence the constant difference between the two.

Below we've included plots of the sum of the averaging kernel as you recommended, and also the averaging kernel function at selected heights in the boundary layer for each retrieval methodology.

We have rephrased the line in question to the following:
Lines 353 to 354: [" In fact, the 10 m - 0 m lapse rate is essentially constant in the MWRoe, implying that retrieved temperatures at 0 m and 10 m are highly correlated."]

[Figure]

[Figure]

*Fig. 8: Please specify the extent of the central box in the legend*

The box plot whiskers in figure 8 extend to include 90% of all data points. The central part of the box plot shows the interquartile range (25$^{th}$-75$^{th}$ percentile). We have included this in the figure caption.

*Appendix B: Can you also show the corresponding figure for the 0-10m lapse rate with and without surface temperature. I would suggest to use this instead of Fig. 6*

We have added this to figure 6. Although we would like to highlight that this is not really a fair comparison since we used the surface tower measurement to constrain the retrievals for the MWRoe-sfc, so the two variables are not independent but, based on your suggestions, we have included it for reference and I make a note of this in the text:

Lines 357 to 361: ["When the in-situ surface temperatures are used to constrain the MWR retrieval (in the MWRoe-sfc), the ability of the retrieval to capture the shallow temperature inversions is considerably improved (Fig. 6a). Note that the correlation between the MWRoe-sfc near surface temperature inversion and the in-situ measurements in Fig. 6a is not a fair assessment of performance since the retrieval results are not independent from the in-situ measurements. Nonetheless, it highlights the importance of using accurate surface temperature measurements to constrain MWR temperature retrievals."]

*- l. 388: ... Results of Appendix B... --> you probably wanted to add a reference to Appendix B*

This line has been deleted since we have integrated appendix B into the main text.

*- l. 427: "AERI contain more information about the temperature near the surface than the MWR measurements". you might add something like "what indicates the importance of including surface observations to the retrieval if available from the MWR's weather station"*

Thank you, we have added this in line 495:

["This highlights the importance of using accurate surface temperature measurements to constrain MWR thermodynamic profile retrievals."]

---

## Author Comment (AC2)

**AMT-2022-12: Response to reviewers**

03 May 2022

**The co-authors would like to thank all reviewers for their feedback and thoughtful suggestions. We have responded to each comment below. All line numbers refer to lines in the revised document unless otherwise stated.**

**In addition to edits made in response to reviewer comments, we have made the following additional minor corrections which have no further implications on the results:**

1) We have corrected a typo on line 201, the number of verified clear sky hours between June and September 2019 is 179 not 236.
2) We have corrected an error in Fig. 4 panel (d): The original plot showed the temperature standard deviation ratio on the x-axis rather than the water vapor standard deviation ratio.

Original reviewer comments are included in *italics* below. Co-author responses are in simple text, and quotations from the revised manuscript are enclosed in square brackets ["..".].

**Response to RC2:**

*General comments:*

*I found the paper interesting and well written, and I recommend it for publication. The figures are of good quality and the presentation is clear. If I may put forward a general comment for a minor modification, in my opinion the discussion part would benefit from a more balanced approach where, instead of trying to prove that the AERI is "better" than the MWR to detect radiation fog (which was not so clear to me at the end of the discussion), the advantages of using both instruments to improve fog forecast under a broad range of conditions are discussed.*

This is a point made be multiple reviewers and we have tried to re-word the entire manuscript to address this. Notable additions include the following:

To the discussion (lines 449 to 469):

["This study focuses on cases of thin radiative fog (LWP < 40 g m-2), which is the most common type of fog at Summit, and draws attention to the benefits of the AERI, which is particularly sensitive to the small changes in LWP and strong shallow temperature inversions that are characteristic of these events. For other types of fog, onset might not be initiated by a small increase in LWP; for example in stratus lowering events, the reduction in cloud base height from the ceilometer might be a better indicator of fog onset. At other locations (in the mid-latitudes for example) thicker fogs with LWP > 50 g m-2 are more common and can be 100's of meters deep (Toledo et al., 2021). Although the AERI might still be a useful instrument for the early detection of such events, once the fog becomes optically thick in the infrared, the AERI can no longer provide information about the thermodynamic profile above

the fog or the trend in LWP, both of which are useful parameters for understanding the development of deep well-mixed fog (Toledo et al.,2021). In such cases, thermodynamic profile and LWP retrievals from the MWR are valuable. The TROPoe algorithm can combine both AERI and MWR measurements in the same retrieval. Below cloud thermodynamic profiles from the combined MWR+AERI are essentially the same as retrievals based on AERI measurements alone (Turner and Lohnert, 2021) but the uncertainty in the LWP retrieval when both instruments are combined is < 20% across the entire range in LWP from 1 to over 500 g m-2 (Turner 2007b).

Although this study focuses on the passive remote sensing instruments that are essential for fog detection (since the active remote-sensing instruments have a blind spot immediately above the surface). Complementary information from active remote-sensing instruments are also necessary for accurate results. We demonstrate in section 3.1 that accurate cloud base height detection (from the ceilometer) is an important input for the AERIoe retrievals, and the radar is also required to filter out precipitation events than can invalidate retrievals from both the MWR and the AERI. Overall, this study highlights the importance of instrument synergy to provide optimal thermodynamic profile and LWP retrievals, supporting the findings of previous studies (Turner et al., 2007a; Löhnert et al., 2009; Turner and Lohnert, 2021; Smith et al., 2021; Djalalova et al.,2021), and expanding on this conclusion to include the specific conditions pertaining to the development of radiation fog."]

And to the conclusion (lines 517 to 521):

["This highlights the importance of a multi-instrument approach to improve fog forecasting under all sky conditions: ceilometer cloud base heights are necessary to generate accurate thermodynamic profile retrievals from the AERI, MWRs are needed to retrieve LWP and thermodynamic profiles above optically thick fog / clouds, and radar data is required to determine the presence of precipitation, which can invalidate retrievals from both passive instruments."]

*To explain better this point, I'll mention here the 3 criteria used in the paper: Accurate temperature and humidity retrieval, characterization of shallow surface inversions, detection of small changes in LWP.*

*The temperature and humidity retrievals (section 3.2), at least in the dataset analyzed, appear reasonable from both instruments. I agree that the bias in the MWR channel is an issue that needs to be addressed, even in assimilation, but it can be addressed. The problem of detecting surface inversions (0-10 m) from the MWR (section 3.3) is easily overcome with the help of surface temperature measurements. Obviously, if using only brightness temperatures, the MWR won't be able to resolve the two heights because measurements at 0 and 10 m are highly correlated (that's why you get a constant difference in Fig. 6a). However, that is the reason why the instrument is equipped with surface sensors.*

To emphasize the importance of including surface temperature measurements in the MWR retrievals and discuss the impact this has on the ability of the MWR retrieval to resolve surface temperature inversions, we have replaced Fig. 5 with Fig. B1 and integrated the discussion from the appendix to the main text.

See lines 225 to 233:

["Due to the limited vertical resolution of the MWR, operational retrievals of thermodynamic profiles from MWRs are typically also constrained by an in-situ measurement of surface temperature, usually from a sensor that is integrated with the MWR (e.g. Cimini et al., 2015). For this study we run TROPoe in three physically consistent configurations; once using only the PAERI radiances as the observation vector (as in Turner et al., 2014, henceforth named AERIoe), once using only the microwave brightness temperature observations from the HATPRO MWR (as in Lohnert et al., 2009) to provide a direct comparison of the relative sensitivity of the two instruments (henceforth named MWRoe), and the final configuration is the same as the MWRoe but is additionally constrained by the in-situ surface temperature and water vapor observations from the HMP155 as it would be in an operational setting (henceforth named MWRoe-sfc)."]

*It therefore appears that the only real advantage of the IR measurements is the higher sensitivity to small changes in LWP (section 3.4). Even here however, this advantage is true in very controlled conditions such as those carefully selected for this specific comparison, i.e. when the sky is clear, no ice, LWP < 40 g/m$^2$, and the cloud (fog) base is accurately known (as shown is section 3.1). It is not entirely obvious to me that a correct identification of all these conditions is easily achieved in NWP models and how the mischaracterization of any of these conditions could affect the results. When everything is put together therefore it almost seems that the advantages of using IR are not that clear. I understand that this paper is an initial step and that a more holistic assessment of fog detection with both instruments is out of its scope, but perhaps some ideas on how to use the synergy of both instruments to obviate shortcomings in each single one would be useful. For example, could having both estimates of fog (MW and IR) help identify cases when one of the two is not acceptable for any reason? Or to identify false positives or missed cases?*

The TROPoe algorithm can combine both AERI and MWR measurements in the same retrieval. Below cloud thermodynamic profiles from the combined MWR+AERI are essentially the same as retrievals based on AERI measurements alone (Turner and Lohnert, 2021) but the uncertainty in the LWP retrieval when both instruments are combined is < 20% across the entire range in LWP from 1 to over 500 g m-2 (Turner 2007b).

We have expanded our discussion of instrument synergy in the additions to the discussion and conclusion section listed above (lines 449 to 469 and lines 517 to 521, quoted above).

---

## Author Comment (AC3)

**AMT-2022-12: Response to reviewers**

03 May 2022

**The co-authors would like to thank all reviewers for their feedback and thoughtful suggestions. We have responded to each comment below. All line numbers refer to lines in the revised document unless otherwise stated.**

**In addition to edits made in response to reviewer comments, we have made the following additional minor corrections which have no further implications on the results:**

1) We have corrected a typo on line 201, the number of verified clear sky hours between June and September 2019 is 179 not 236.
2) We have corrected an error in Fig. 4 panel (d): The original plot showed the temperature standard deviation ratio on the x-axis rather than the water vapor standard deviation ratio.

Original reviewer comments are included in *italics* below. Co-author responses are in simple text, and quotations from the revised manuscript are enclosed in square brackets ["..."].

**Response to RC3:**

*Overview:*

*This paper investigates the capability of two passive instruments, the AERI and HATPRO MWR to detect shallow surface-based temperature inversions and to provide good retrievals of LWP under thin radiative fog events. I think the paper is very well written and of a very good scientific quality. It is also very interesting and important for the scientific community as this is the first time the benefit of the AERI instrument for fog forecast is evaluated. However, there are a few points that I think are important to clarify before the publication of this manuscript.*

*I would recommend a more moderated conclusion through the manuscript instead of trying to prove that AERI is better than the MWR for fog forecast improvements.*

This is a point made by multiple reviewers and we have tried to re-word the manuscript to address this. Notable additions include the follow:

To the discussion (lines 449 to 469):

["This study focuses on cases of thin radiative fog (LWP < 40 g m-2), which is the most common type of fog at Summit, and draws attention to the benefits of the AERI, which is particularly sensitive to the small changes in LWP and strong shallow temperature inversions that are characteristic of these events. For other types of fog, onset might not be initiated by a small increase in LWP, for example in stratus lowering events, the reduction in cloud base height from the ceilometer might be a better indicator of fog onset. At other locations (in the mid-latitudes for example) thicker fogs with LWP > 50 g m-2 are more common and can be

100's of meters deep (Toledo et al., 2021). Although the AERI might still be a useful instrument for the early detection of such events, once the fog becomes optically thick in the infrared, the AERI can no longer provide information about the thermodynamic profile above the fog or the trend in LWP, both of which are useful parameters for understanding the development of deep well-mixed fog (Toledo et al.,2021). In such cases, thermodynamic profile and LWP retrievals from the MWR are valuable. The TROPoe algorithm can combine both AERI and MWR measurements in the same retrieval. Below cloud thermodynamic profiles from the combined MWR+AERI are essentially the same as retrievals based on AERI measurements alone (Turner and Lohnert, 2021) but the uncertainty in the LWP retrieval when both instruments are combined is $< 20\%$ across the entire range in LWP from 1 to over 500 g m-2 (Turner 2007b).

Although this study focuses on the passive remote sensing instruments that are essential for fog detection (since the active remote-sensing instruments have a blind spot immediately above the surface). Complementary information from active remote-sensing instruments are also necessary for accurate results. We demonstrate in section 3.1 that accurate cloud base height detection (from the ceilometer) is an important input for the AERIoe retrievals, and the radar is also required to filter out precipitation events than can invalidate retrievals from both the MWR and the AERI. Overall, this study highlights the importance of instrument synergy to provide optimal thermodynamic profile and LWP retrievals, supporting the findings of previous studies (Turner et al., 2007a; Löhnert et al., 2009; Turner and Lohnert, 2021; Smith et al., 2021; Djalalova et al.,2021), and expanding on this conclusion to include the specific conditions pertaining to the development of radiation fog."]

And to the conclusion (lines 517 to 521):

["This highlights the importance of a multi-instrument approach to improve fog forecasting under all sky conditions: ceilometer cloud base heights are necessary to generate accurate thermodynamic profile retrievals from the AERI, MWRs are needed to retrieve LWP and thermodynamic profiles above optically thick fog / clouds, and radar data is required to determine the presence of precipitation, which can invalidate retrievals from both passive instruments."]

*In fact, I think the major issue of the lack of visibility data should really limit the conclusions that AERI can really detect fog onset before the MWR. To be more objective, with the current dataset used in the paper, what is demonstrated is an increase signal in LWP detected earlier with AERI compared to the MWR but without « real » proof that this is related to fog presence.*

This somewhat comes down to the chosen definition of fog. For many practical purposes, fog is defined by a visibility threshold (normally < 1,000 m horizontal visibility). Using this definition, we agree that it is not possible to determine the ability of either instrument to detect fog onset in the absence of continuous horizontal visibility data. However, neither instrument is suitable for the direct measurement of horizontal visibility (which is a function of fog depth and particle size distribution in addition to LWP), and this is not what we are trying to demonstrate in this analysis.

Rather than focusing on horizontal visibility, we focus on the detection of the presence of near surface liquid water, which is a requirement for fog, and is additionally important from a radiative perspective, and for moisture/aerosol cycling in the boundary layer. Limiting the

definition of fogs to only those which reduce visibility to less than 1,000 m, although practical for safety concerns, encourages thinner fogs to be dismissed or incorrectly classified as clear sky events. Being able to accurately measure these thinner fog events is extremely important because (a) they form the precursor to thicker fog and hence can potentially provide an early warning, (b) even if they do not develop into thicker fog, they modify the surface moisture, aerosol, temperature and radiative structure which might impact fog development later down the line, and (c) they can have important radiative and climatological influences even without developing into a thick fog, but are not captured well in numerical models, so accurate measurements are essential for improving model performance in this area.

We have tried to make this point clearer by adding the following paragraph at the beginning of section 2.2:

Lines 184 to 192:
["For forecasting and nowcasting purposes, fog is usually defined by a threshold in horizontal visibility (typically < 1,000 m) which has important implications from a safety perspective (Gultepe et al., 2007). However, limiting the definition of fogs to those that reduce visibility to < 1,000 m encourages thinner fogs (or mists) to be ignored or incorrectly classified as clear sky events. Being able to accurately measure thinner fogs is extremely important because (a) they form the precursor to thick fog, (b) they modify the surface moisture, aerosol, temperature and radiative structure which might impact fog development further down the line (Haeffelin at al., 2013) and (c) they can have important radiative and climatological impacts even without developing into a thick fog (Cox et al., 2019, Hachfeld et al., 2000). Because both the MWR and AERI are inherently sensitive to the radiative impact of fog (as opposed to visibility), for the purpose of this study, we define fog as the presence of near surface liquid water that has a detectable radiative impact."]

Nonetheless, the fact that our independent 'truth' dataset is limited is still a good point, and we have tried to address this by including more data from the ceilometer (which is sensitive to the presence of liquid water drops above 15 m but not very sensitive to ice crystals, Van Trich et al., 2014). We provide more details of these changes in our response to your specific recommendations below.

*Secondly, I think the discussion on the temperature lapse rate should also be more widely discussed in the paper because in-situ temperature surface from the MWR should be integrated in the 0-10m lapse rate comparisons.*

We have integrated appendix B into the main text and included the lapse rate comparison of the MWRoe-sfc (with the surface temperature constraint) onto Fig. 6. However, it is important to note that when the MWRoe is constrained by the surface temperature measurements, the retrieval results are no longer independent from the tower measurements and the correlation reflects this.

We have pointed this out in the text:
Lines 357 to 361: ["When the in-situ surface temperatures are used to constrain the MWR retrieval (in the MWRoe-sfc), the ability of the retrieval to capture the shallow temperature inversions is considerably improved (Fig. 6a). Note that the correlation between the MWRoe-sfc near surface temperature inversion and the in-situ measurements in Fig. 6a is not a fair assessment of performance since the retrieval results are not independent from the in-situ

measurements. Nonetheless, it highlights the importance of using accurate surface temperature measurements to constrain MWR temperature retrievals."]

*I also believe that the temperature inversion over a thicker layer might already be a good proxy for radiative fog and the comparison considering only a 10m thick layer is clearly penalizing the MWR.*

The comparison of the 100 m – 10 m lapse rate between the retrievals and radiosonde profiles for the 14 coincident radiosonde launches has been added to Fig. 6. We disagree that showing the 10 m – 0 m  comparison is "penalising" the MWR, rather it is demonstrating the true limitations of the instrument.

*Finally, I think a discussion about the use of LWP retrievals for fog nowcasting dissipation (Toledo et al 2021, ACP) would be beneficial to the article. In fact, even if AERI might detect fog formation a bit earlier due to very low LWP values, what about potential limitations for fog dissipation when the fog is thicker and might reach the saturation signal of the AERI ? Would the AERI be a good candidate to apply the conceptual model described in Toledo et al 2021 or the MWR would be a better candidate this time ? This question might probably lead to a more balanced conclusion not trying to put the AERI against the MWR but to open the perspective of the instrumental synergy highlighting the benefit of each instrument depending on the fog stages.*

Toledo et al., (2021) focus on the dissipation of well mixed adiabatic fog (with LWP > 30-40 g m$^{-2}$ and with fog top heights > 85 m) through a reduction in LWP and/or fog base lifting. They develop a parameterisation to determine the critical LWP required to sustain visibility < 1,000 m at the surface for a given fog/cloud top height.  According to this parameterization, when fog top heights are greater than 250-300 m, the critical LWP required to sustain fog is greater than 50 g m$^{-3}$. In such cases, when the fog is opaque in the infrared, the AERI would not be sensitive to the changes in LWP that would indicate fog dissipation and the MWR LWP would be required. For fogs with cloud tops lower than 200 m, the critical LWP is < 40 g m$^{-2}$, and on these such occasions, a more accurate retrieval of LWP from an instrument such as the AERI could potentially be useful. When AERI and MWR measurements are combined in the TROPoe algorithm (Turner 2007b, Turner and Lohnert, 2021), the uncertainty in the LWP retrievals is < 20% across the entire range of LWP (1 to < 500 g m-2), which is potentially the optimal solution (with the disadvantage that it requires two instruments).

None of the cases we consider in this study are suitable candidates for this algorithm, since the LWP is almost always < 40 g m$^{-2}$ and the fogs rarely become well mixed – this is a feature of the very dry and shallow boundary layers at Summit. We realise that by not including thicker fogs we are neglecting to discuss a whole category of fogs that might be particularly relevant (and more impactful) at mid-latitudes.

In the introduction and abstract we have drawn attention to the fact that we are only focusing on radiatively thin fogs, and to the discussion we have added the following paragraph:

To the discussion (lines 449 to 469):

["This study focuses on cases of thin radiative fog (LWP < 40 g m-2), which is the most common type of fog at Summit, and draws attention to the benefits of the AERI, which is

particularly sensitive to the small changes in LWP and strong shallow temperature inversions that are characteristic of these events. For other types of fog, onset might not be initiated by a small increase in LWP, for example in stratus lowering events, the reduction in cloud base height from the ceilometer might be a better indicator of fog onset. At other locations (in the mid-latitudes for example) thicker fogs with LWP > 50 g m-2 are more common and can be 100's of meters deep (Toledo et al., 2021). Although the AERI might still be a useful instrument for the early detection of such events, once the fog becomes optically thick in the infrared, the AERI can no longer provide information about the thermodynamic profile above the fog or the trend in LWP, both of which are useful parameters for understanding the development of deep well-mixed fog (Toledo et al.,2021). In such cases, thermodynamic profile and LWP retrievals from the MWR are valuable. The TROPoe algorithm can combine both AERI and MWR measurements in the same retrieval. Below cloud thermodynamic profiles from the combined MWR+AERI are essentially the same as retrievals based on AERI measurements alone (Turner and Lohnert, 2021) but the uncertainty in the LWP retrieval when both instruments are combined is < 20% across the entire range in LWP from 1 to over 500 g m-2 (Turner 2007b).

Although this study focuses on the passive remote sensing instruments that are essential for fog detection (since the active remote-sensing instruments have a blind spot immediately above the surface). Complementary information from active remote-sensing instruments are also necessary for accurate results. We demonstrate in section 3.1 that accurate cloud base height detection (from the ceilometer) is an important input for the AERIoe retrievals, and the radar is also required to filter out precipitation events than can invalidate retrievals from both the MWR and the AERI. Overall, this study highlights the importance of instrument synergy to provide optimal thermodynamic profile and LWP retrievals, supporting the findings of previous studies (Turner et al., 2007a; Löhnert et al., 2009; Turner and Lohnert, 2021; Smith et al., 2021; Djalalova et al.,2021), and expanding on this conclusion to include the specific conditions pertaining to the development of radiation fog."]

*Major points :*

*Lack of visibility data :*

*One of the major issues for me is the lack of visibility measurements which is the « reference » instrument to detect fog events. This is for me especially problematic for figures 10 and 11. I think, lacking this important reference measurements, the authors are going a bit too fast concluding that AERI is able to detect fog onset 4 hours before the MWR. What is true is that we detect a signal of LWP increase in the AERI earlier than observed in the MWR. However, as there is no reference instrument to give the exact time of fog formation, we cannot rely on the fact that an increase of 0.1 g/m² in the AERI LWP determines the true time of fog formation. I think it is particularly important as, if I understood well, the AERI LWP increase could also be due to the presence of ice crystals in the atmosphere that can't be detected by the MWR for example.*

As mentioned above, determining the 'exact time' of fog formation is definition dependent, and in this study our focus is on the presence versus absence of near surface liquid water as an indicator of fog rather than a horizontal visibility criterion. This is partly because of lack of visibility data, but also because the ability to detect even small amounts of liquid water is important (mentioned above).

We have added an independent definition of 'fog onset' from the ceilometer. We used the same subset of verified clear days that we used to identify the fog signal form the AERI initially to determine the distribution of 'clear sky' total backscatter from the ceilometer and identified fog onset from the ceilometer as the time when the total backscatter increases greater than 3 standard deviations above the mean clear sky total backscatter. The ceilometer backscatter is insensitive to low concentrations of ice crystals (Van Tricht et al., 2014), and the ceilometer also detects fog in 11/12 of the events, suggesting that these fogs were indeed present but not well detected by the MWR.

We have described this addition in lines 404 to 408:

["For independent verification, we also determine fog onset from the ceilometer range-corrected attenuated backscatter. We define the ceilometer fog onset as where the 5-minute mean total backscatter increases greater than three standard deviations from the mean clear sky backscatter at Summit between 01 June and 30 September 2019 (the mean clear sky backscatter is determined using the same subset of verified clear sky hours used to identify fog events from the AERI radiance, section 2.2)"]

We have added the ceilometer derived 'fog onset' to figure 11 for comparison to the other two instruments and described the results in lines: 410 to 416 (below). I also realised in inspecting this closely that some of the MWRoe fog onset times occurred after the end of the 'fog event' defined in Table 3 and were actually a result of low-level cloud moving in after the end of the fog event (this was the case for the 5$^{th}$ Sept and 8$^{th}$ June case). We have now restricted the fog onset determination to the actual fog time window listed in the updated Table 3 (rather than the extended time for which we ran the TROPoe). This has resulted in a slight change in the values depicted in figure 11 and we have updated the text accordingly (see below).

Lines 412 to 418:

["The ceilometer detects fog for all cases with the exception of case 7 (04 August). During this case the fog was extremely thin (maximum LWP from the AERI only 2 g m−2), but the onsite observer logged the presence of a fog bow between 07:15 and 08:30, demonstrating that liquid water droplets were indeed present. This was a very marginal case that demonstrates the ability of the AERI to detect very small amounts of liquid water when even the ceilometer cannot. The MWRoe retrieval only detects fog for 6/12 cases (Fig. 11), and for those 6 cases, the AERIoe retrieval consistently detects the onset of fog (via the increase in LWP) before the MWRoe retrieval by 25 to 185 minutes (Fig. 11). For the 6 cases where the MWRoe does not detect the fog, the mean LWP detected by the AERIoe is very low (1.4 to 3.1 g m−2)"]

*To help clarifying this point, can the authors provide with figure 10, the time series of ceilometer CBH / vertical visibility for that specific day and the time serie of relative humidity observed at the ground and 10m on the tower ? In fact, I would expect even thin radiative fogs to be detected by the ceilometer which should provide a CBH lower than 50m within these conditions or a vertical visibility and could be a first good validation of when the fog really forms. You could also look at the time series of relative humidity (RH) from the tower measurements : though a RH > 95 % is not always a good proxy of the presence of fog, a RH < 95 % would easily show that there is little chance of fog.*

We have added the time series of ceilometer total backscatter and vertical visibility (note that no CBH were reported by the ceilometer) this event to figure 10.

As for the relative humidity measurements, the near surface RH profile measured at Summit is quite interesting in that we rarely measure 100% RH even when we know there is fog present (i.e. surface visibility is reduced to < 1,000 m). Possibly this is a calibration issue with the sensors, or perhaps it is a unique property of the environment at Summit where the fog droplets form in a saturated layer 10's of m above the surface and settle/ evaporate in the very near surface layer. There is some evidence for the latter in the work of Cox et al., (2019) and Berkelhammer et al., (2016). It is outside the scope of this study to investigate this conundrum further, so we choose not to show the relative humidity measurements in this study as we believe they will add a layer of confusion and take away from the main message without adding additional value. However, we do include a plot of the RH measurements for the 15 July case study below for your reference. The maximum 10 m RH occurs at the same time as fog onset is detected by the AERIoe, and the RH is higher at 10 m than at the surface, providing support to the idea that the fog droplets form in a saturated layer above this height. In any case the vertical visibility at 00h was only 400 m, indicating that fog was present despite the apparently unsaturated surface layer.

[Figure]

*I think this lack of visibility data might also be problematic for the definition of the 13 fog cases provided in table 3. I understand that only two criteria have been used : an increase in the downwelling infrared radiances representative of a cloud presence and no cloud detected above 200m by the MMCR. However, as mentioned later in the manuscrit, the ceilometer CBH should be used to detect clouds with CBH below 50m or ceilometer data providing a vertical visibility in general informing of fog presence to at least avoid wrongly classifying low clouds (with a CBH < 200 m) in fog if they do not reach the ground.*
*It would be interesting to specify for each of the 13 fog events identified the maximal CBH or vertical visibility height within each fog event detected by the CT25K to be sure that no potentially « low louds » have been wrongly classified as fog.*

We have added the minimum ceilometer vertical visibility field to Table 3 to give a further independent indication of the visibility reduction at the surface during each event. Except for event ID 12, the ceilometer does not detect a cloud base height during any of these events, so rather than add an additional column to the table, we added this the following statement:

Lines 207 to 210 :
["Note that for 11 of these cases, there is no cloud base height detected by the ceilometer during the event indicating that the events were indeed fog as opposed to low cloud. The only exception is for case ID 11, during which the ceilometer detected a cloud base between 52 and 105 m intermittently between periods of obscured vertical visibility."]

We choose to retain event 11 even though the fog appears to intermittently lift from the surface since during most of the event the ceilometer does report restricted vertical visibility and on-site observers logged FZFG on that day.

*The current methodology used in the paper is particularly questioning when looking at figure 3 which shows the ceilometer CBH for an identified radiative fog starting at 2 UTC and ending at 12 UTC. Here the ceilometer CBH is around 1300 m until 3h30 while table 3 specifies a starting fog time at 2 UTC : could you explain how fog could form even in the presence of this low cloud and why the MMCR does not detect any cloud at that altitude ?*

This point arises from the fact that after the identification of events, we expanded the time either side of the events during which we would run the TROPoe retrievals so that we could include the conditions before fog onset and after the fog dissipation. We realise that it was misleading to include the times for which we ran the TROPoe in Table 3 rather than the start and end times of the fog as per our initial case identification criteria. We have updated the times in Table 3 to correct this and we have updated Figure 2 to reflect the corrected times. We added Line 251 to clarify that the TROPoe runs encapsulated the times prior to fog onset and after dissipation.

*The definition of the fog events used in this study is also questionable when it is mentioned line 223 that, for some fog events, no CBH or vertical visibility is provided by the ceilometer : in that case how can we be sure that the increase in downwelling infrared radiances is not due to ice particles instead of fog presence that would not be detected by the ceilometer ?*

Part of this is due to the fact mentioned above, that we expanded the TROPoe runs either side of the fog event to capture the onset and dissipation, so there are times included in the retrievals where the ceilometer does not see any fog.

Aside from that, there were also three cases where the ceilometer does not report obscured vertical visibility during the detected 'fog'. One of these cases (the 9[th] August case), is an extremely tenuous case and (although the observer reported a horizontal visibility reduction to 1,600 m) I agree that it is not possible to determine whether or not the signal is due to liquid droplets or ice crystals. We have therefore removed this case from the study (and updated all plots accordingly). The other two cases you can see in the updated Table 3 are cases IDs 7 and 12.

For case 7, the total ceilometer backscatter is never high enough to indicate the presence of liquid water. However, the onsite observer logged the presence of a fog bow between 07:15 and 08:30 that is indicative of liquid water droplets (ice particles do not form fog bows). The fog bow was also visible in TSI images from the 04 Aug 2019 (see below). Clearly this was a very marginal case that demonstrates the ability of the AERI to detect very small amounts of liquid water when even the ceilometer cannot.

TSI images from 07:00 and 08:00 UTC on 04 August 2019:

[Figure]

For case 13, although the ceilometer does not report obstructed vertical visibility, the signal is clearly attenuated (see the backscatter plot below), and the ceilometer still detects a 'fog onset' based on the criteria described in lines 402 to 405. The signal in the AERI in this case is equivalent to a maximum liquid water path of 4.8 g m$^{-2}$. If the equivalent optical depth resulted from ice crystals, these would likely have been observed in the POSS. The onsite observer log verifies FZFG.

Ceilometer backscatter for 30$^{th}$ September 2019:

[Figure]

*Temperature lapse rate :*

*It is well known that MWRs cannot provide two independent informations of temperature at surface and 10m a.g.l as their vertical resolution is approximately higher than 50m. This is a common approach to combine the MWR with an in-situ surface station (either the one provided with the HATPRO or an external station as MWRs are often deployed in instrumented sites). This combination is used for atmospheric boundary layer height detection and the detection of stable boundary layer conditions from MWR measurements often based on the temperature difference between a higher altitude level and 50m (and not the MWR retrievals at 0m). In that sense, I think figure 6 is not entirely objective in the way the MWR lapse rate is evaluated and compared against the AERI. However, this figure demonstrates the capability of the AERI to retrieve this lapse rate while the MWR can't without integrating the surface station. For a more balanced conclusion, I would first recommend to include in figure 6, the comparison with the*

*MWR lapse rate calculated with the in-situ surface temperature station integrated with the MWR (as it is shown in Appendix B).*

We have included the lapse rate comparison of the MWRoe-sfc (with the surface temperature constraint) onto Fig. 6. However, we think it is important to note that when the MWRoe is constrained by the surface temperature measurements, the retrieval results are no longer independent from the tower measurements and the correlation reflects this.

We have pointed this out in the text:
Lines 357 to 361: ["When the in-situ surface temperatures are used to constrain the MWR retrieval (in the MWRoe-sfc), the ability of the retrieval to capture the shallow temperature inversions is considerably improved (Fig. 6a). Note that the correlation between the MWRoe-sfc near surface temperature inversion and the in-situ measurements in Fig. 6a is not a fair assessment of performance since the retrieval results are not independent from the in-situ measurements. Nonetheless, it highlights the importance of using accurate surface temperature measurements to constrain MWR temperature retrievals."]

*Additionally, I think the authors should also compare the AERI lapse rate and MWR lapse rates together with higher altitude levels (like T100m-T50m). This is important because even if the figure demonstrates the capability of the AERI to better capture the 0-10m temperature inversion, I am not entirely convinced that the 0-10m lapse rate is the « key » proxy of fog formation. I expect temperature inversions during radiative fog to spread over a larger atmospheric layer, where there is a high chance that the MWR can detect the inversion as well as the AERI and could already be an information sufficient to improve fog nowcasting. As for data assimilation, an improved lapse rate between 0 and 10m seems too resolved compared to the capability of current data assimilation systems (and knowing that surface stations are already assimilated in NWP models, I think it is really pertinent to investigate the MWR capability versus AERI on a larger layer than only 10 m…).*

We have also added a comparison of the 100 m – 10 m lapse rate between the retrievals and radiosonde profiles for the 14 coincident radiosonde launches to Fig. 6.
The additional discussion of this figure we have included in the following lines:

Lines 362 to 366: ["The radiosonde profiles provide an alternative independent measure of surface inversion strength, allowing the comparison of the ability of each retrieval configuration to capture surface temperature inversions over a deeper layer. Fig 6b compares the 100 m - 10 m retrieved inversion strength with that measured by the 14 coincident radiosonde profiles. Over this depth the RMSE of the AERIoe and the MWRoe-sfc are comparable to the values for the 10 m - 0 m comparison (1.65 and 1.83 C m$^{-1}$ respectively), but the MWRoe RMSE remains much larger (2.22 C m$^{-1}$), demonstrating that the MWRoe alone is not capable of accurate retrievals of surface temperature inversions even in this deeper layer. Only the AERIoe retrievals in this case are significantly correlated (r=0.46) with the radiosonde measurements, although the small number of radiosondes available for comparison makes it difficult to draw robust conclusions from this result. Klein et al. (2015) compared AERI derived lapse rate 100 m -10 m against more than 200 radiosondes in Oklahome (southern US) , and found very good agreement with r2 values > 0.93 "]

*Minor points :*

*line 84 : large error during thin fogs → large **relative** errors during thin fogs*

I added relative here – thanks!

*line 154 : plus the current and temperature of Stirling cooler : I don't entirely understand the sentence as if one word is missing after « current ».*

I see that this is confusing! I have added the word 'electric' so it's clear that we're monitoring the electric current and temperature of the Stirling cooler.

*line 388 : It looks like « Results of Appendix B » should appear within parenthesis and the « ... » before and after should be removed.*

This was indeed a mistype but has now been removed since Appendix B has been integrated into the main text.

---

## Referee Report (RR1)

First of all, I would like to thank the authors for their efforts to take into account the suggestions and corrections especially for including the discussion with the integration of the in-situ surface sensor within the MWR retrievals and including the ceilometer measurements to better detect fog in the lack of visibility measurements. I am just wondering if what is defined in the manuscript as the fog detection with the ceilometer could be associated with aerosol hygroscopic growth as shown in Haeffelin et al 2016. In that sense I think it would be good to remind the reader of the new fog definition used in the manuscript : not visibility < 1000m as usual but : « the presence of near surface liquid water that has a detectable radiative impact ». I thus suggest some minor corrections following this direction throught the paper. The line numbering is refering to the author's tracked changes document.

Line 90 : Can you mention that the relative LWP uncertainties from MWR for low LWP alone can be reduced by a combination from a 1 channel infrared spectrometer (Marke et al, 2016).

Table 3 is cut on the right.

Line 287 : initial increase in LWP associated with fog formation and visibility reduction. As you cannot be 100 % sure that this is associated with fog I would change into :
« initial increase in LWP defined as an indicator of fog formation in this study that might lead to visibility reduction by defining 'fog onset' as where the retrieved ….. »

Line 409 : as the new fog definition used in the manuscript has been defined quite far from this section, I would remind the reader of this definition here :
the development of radiation fog under clear skies is detected earlier in the AERIoe retrievals compared to the MWRoe (following our fog definition as the presence of near surface liquid water that has a detectable radiative impact).

Line 418 (figure 10) : Could the authors clarify if they are sure that the new « fog » definition from the ceilometer backscatter coefficient (increase in the ceilometer mean backscatter by more than three standard-deviations) represent well the fog formation and not the aerosol hygroscopic growth. In fact, Haeffelin et al 2016 demonstrated that an increase in the ceilometer backscatter coefficients can be associated with the aerosol hygroscopic growth and not the fog formation. This is used in pre-fog alerts to identify the aerosol hygroscopic growth occuring in fact before fog formation. I have some doubts because, in figure 10, the ceilometer output  is « obscurred signal » and not « vertical visibility » up to ~1h30 UTC. This would make a difference with the MWR detection of ~ 40 minutes instead of ~ the announced 2 hours by using the fog onset definition based on the LWP increase or the ceilometer backscatter coefficient increase.
Could you please explicit why you think that the ceilometer backscatter coefficient increase can be considered as fog formation between 0 and 1h30 and not due to aerosol hygroscopic growth ?
If you think that obscurred signal means that droplets are present, in that sense it would be in line with your new fog definition but I think it would be valuable for the paper to just add a few lines of discussion on this subject (aerosol growth versus presence of fog droplets and remind again that if this is for sure liquid droplets this is in line with the fog definition used in the paper).

Line 536 : « This means that the AERIoe is consistently able to detect small changes in LWP that signify the onset of radiation fog and reduction in horizontal visibility ». According to your fog definition, I would change into :

This means that the AERIoe is consistently able to detect small changes in LWP that might initiate radiation fog and reduction in horizontal visibility

---

## Author Response (AR2)

**AMT-2022-12: Response to review of revised submission**

03 August 2022

**We would like to thank the editor and two reviewers for carefully considering our revised manuscript and provide a response to the addition recommendations from reviewer #3 below.**

Original reviewer comments are included in *italics*. Co-author responses are in simple text, and quotations from the revised manuscript are enclosed in square brackets ["..."]. All line numbers refer to the final revised version of the manuscript.

**Response to anonymous referee report #2:**

*Line 90 : Can you mention that the relative LWP uncertainties from MWR for low LWP alone can be reduced by a combination from a 1 channel infrared spectrometer (Marke et al, 2016).*

I have added the following sentence:

Lines 88-90: ["These errors can be reduced by combining the MWR data with measurements from infrared spectrometers (either with single or multiple channels) that are more sensitive to small amounts of liquid water (Marke et al., 2016.)"]

*Table 3 is cut on the right.*

This is only the case for the tracked-changes document, the table is correct for the final submission.

*Line 287 : initial increase in LWP associated with fog formation and visibility reduction. As you cannot be 100 % sure that this is associated with fog I would change into :*
*« initial increase in LWP defined as an indicator of fog formation in this study that might lead to visibility reduction by defining 'fog onset' as where the retrieved ..... »*

I have made this change (line in revised manuscript: 283-284)

*Line 409 : as the new fog definition used in the manuscript has been defined quite far from this section, I would remind the reader of this definition here :*
*the development of radiation fog under clear skies is detected earlier in the AERIoe retrievals compared to the MWRoe (following our fog definition as the presence of near surface liquid water that has a detectable radiative impact).*

I have added this sentence (line in revised manuscript: 401)

*Line 418 (figure 10) : Could the authors clarify if they are sure that the new « fog » definition from the ceilometer backscatter coefficient (increase in the ceilometer mean backscatter by more than three standard-deviations) represent well the fog formation and not the aerosol hygroscopic growth. In fact, Haeffelin et al 2016 demonstrated that an increase in the ceilometer backscatter coefficients can be associated with the aerosol hygroscopic growth and not the fog formation. This is used in pre-fog alerts to identify the aerosol hygroscopic*

*growth occuring in fact before fog formation. I have some doubts because, in figure 10, the ceilometer output is « obscurred signal » and not « vertical visibility » up to ~1h30 UTC. This would make a difference with the MWR detection of ~ 40 minutes instead of ~ the announced 2 hours by using the fog onset definition based on the LWP increase or the ceilometer backscatter coefficient increase.*

*Could you please explicit why you think that the ceilometer backscatter coefficient increase can be considered as fog formation between 0 and 1h30 and not due to aerosol hygroscopic growth ?*

*If you think that obscured signal means that droplets are present, in that sense it would be in line with your new fog definition but I think it would be valuable for the paper to just add a few lines of discussion on this subject (aerosol growth versus presence of fog droplets and remind again that if this is for sure liquid droplets this is in line with the fog definition used in the paper).*

Thank you for highlighting this point. We do not distinguish between the detection of droplet formation or aerosol hygroscopic growth using the simple ceilometer backscatter threshold and I have added the following sentences to make this clear:

Lines 412 to 418: ["Ceilometer attenuated backscatter is sensitive the scattering cross section or molecules and particles in the atmosphere and can be sensitive the presence of atmospheric aerosols (e.g. Markowicz et al., 2008), and to the hygroscopic growth of aerosols prior to their activation into fog droplets (Haeffelin et al., 2016), the latter of which can be a precursor to radiation fog formation (Haeffelin et al., 2016). At Summit, the aerosol scattering cross section is usually extremely small ($< 2\text{x}10^{-6}$ m$^{-1}$ at 550 nm, Schmeisser et al., 2018), and any signal due to the presence of aerosols is incorporated into the calculation into the mean 'clear-sky' backscatter. We do not distinguish between the detection of aerosol hygroscopic growth and droplet formation in the ceilometer backscatter"]

For the case study described in the text and shown in figure 10, we can say for certain that that there was fog present reducing visibility to just 400 m at 00 UTC since it was recorded by the onsite observer performing weather observations at the time (described in lines 420 to 421). At this time the ceilometer did only report an 'obscured' signal (rather than vertical visibility, fig. 10) demonstrating that this can indeed be the case when fog is present (even using the more traditional definition of fog).

I also added in line 424 the ceilometer signal could be either due to droplet formation or aerosol hygroscopic growth.

*Line 536 : « This means that the AERIoe is consistently able to detect small changes in LWP that signify the onset of radiation fog and reduction in horizontal visibility ». According to your fog definition, I would change into :*

*This means that the AERIoe is consistently able to detect small changes in LWP that might initiate radiation fog and reduction in horizontal visibility*

I have made this change (line in revised manuscript: 517)